# Imbalance-Regularized LoRA: A Plug-and-Play Method for Improving Fine-Tuning of Foundation Models

## Abstract

Low-Rank Adaptation (LoRA) is an effective fine-tuning algorithm for large models, enabling efficient adaptation with fewer trainable parameters. Despite its success, there remains significant potential for improving LoRA's performance. In this paper, we introduce **iLoRA** (Imbalance-Regularized LoRA), which enhances LoRA by incorporating a regularization term to capture the imbalance in forward propagation. This regularization maintains an imbalance between matrices $\mathbf{A}$ and $\mathbf{B}$, ensuring stable activation variance independent of dimension. Specifically, we first analyze forward dynamics, observe this imbalance in stable training, and introduce imbalanced regularization. Further, by combining this with preconditioning techniques (Zhang and Pilanci, 2024), we propose $\pi$**LoRA** (Preconditioned iLoRA), which improves the backpropagation process. Our method is a plug-and-play algorithm that requires only minor modifications to the existing code and incurs negligible additional computational overhead. Finally, experiments on large language models and text-to-image models demonstrate that iLoRA and $\pi$LoRA significantly outperform existing LoRA and preconditioned LoRA methods.

## 1 Introduction

As neural network models in both vision and language domains continue to grow, training a neural network from scratch to match the performance of existing large models has become increasingly difficult (Brown et al., 2020; Fedus et al., 2022; Zhai et al., 2022; Dubey et al., 2024). Consequently, fine-tuning has emerged as a popular approach for downstream tasks (Devlin, 2018; Liu, 2019). Traditional full-parameter fine-tuning requires extensive storage, making it impractical for many applications (Raffel et al., 2020). In contrast, recent advances in Parameter-Efficient Fine-Tuning (PEFT) methods offer a more efficient solution while maintaining strong performance in downstream tasks (Houlsby et al., 2019; Lester et al., 2021; Hu et al., 2022; Zhang et al., 2023; Hayou et al., 2024; Zhang and Pilanci, 2024; Tian et al., 2024; Zhu et al., 2024; Dettmers et al., 2024).

One widely used PEFT method is Low-Rank Adaptation (LoRA) (Hu et al., 2022), which introduces low-rank matrices to existing model weights and only trains these additive components. Specifically, for a pre-trained weight $\mathbf{W}^{(0)} \in \mathbb{R}^{m \times n}$, LoRA assumes that the fine-tuned weight $\mathbf{W}^*$ satisfies:

$$\mathbf{W}^{\star} = \mathbf{W}^{(0)} + \Delta\mathbf{W} = \mathbf{W}^{(0)} + \mathbf{B}\mathbf{A},$$

where $\mathbf{A} \in \mathbb{R}^{r \times n}$ and $\mathbf{B} \in \mathbb{R}^{m \times r}$, with $r \ll \{m, n\}$ representing the rank of the adaptation matrices. During fine-tuning, only $\mathbf{A}$ and $\mathbf{B}$ are updated, while the original pre-trained weights $\mathbf{W}^{(0)}$ remain frozen. This low-rank factorization significantly reduces memory and computational overhead (Sainath et al., 2013), as the rank $r$ is chosen to be much smaller than the dimensions of $\mathbf{W}^{(0)}$; for example, $r = 8$ when $m = n = 1024$. Despite introducing fewer than 2% additional trainable parameters, LoRA achieves comparable, and sometimes even better, performance than full parameters fine-tuning (Hu et al., 2022). Additionally, the multiplication of two small matrices is more efficient and easier to implement in practice compared to the unstructured sparse matrices used in methods like Diff-pruning (Fang et al., 2023a;b), making LoRA a practical and scalable solution for fine-tuning large models.

LoRA+ (Hayou et al., 2024) further examined the optimization paradigm of LoRA, revealing that for stable feature learning, the learning rate of the parameter $\mathbf{B}$ should be set larger than that of

**A**, leading to a joint hyperparameter search problem. LoRA+ proposed using a heuristic learning rate ratio $\eta_{\mathbf{B}}/\eta_{\mathbf{A}} = 2^4$, and only $\eta_{\mathbf{A}}$ is tuned in practice. Additionally, Riemannian Preconditioned LoRA (Zhang and Pilanci, 2024) introduced a $r \times r$ preconditioner in each gradient step to stabilize feature learning without requiring different learning rates. This method is based on a novel Riemannian metric (Absil et al., 2008) and has proven effective in low-rank matrix estimation (Tong et al., 2021b;c; Zhang et al., 2024; Jia et al., 2023). However, both approaches have notable limitations. The heuristic learning rate ratio in LoRA+ ($\eta_{\mathbf{B}}/\eta_{\mathbf{A}} = 2^4$) does not fully align with their theoretical analysis and overlooks the dimension of matrices **A** and **B**, particularly in models with large dimensions. While Preconditioned LoRA stabilizes feature learning, it does not capture the inherent asymmetry between **A** and **B**. This asymmetry is critical for maintaining consistent forward and backward propagation dynamics and for ensuring efficient utilization of the parameter space. As a result, these methods can lead to suboptimal convergence and propagation behaviors in complex, large-scale models.

In this paper, we introduce **Imbalance-Regularized LoRA (iLoRA)**, a novel method to capture the inherent asymmetry between matrices **A** and **B**. Drawing on insights from the low-rank matrix factorization literature (Tu et al., 2016; Zhu et al., 2021), where regularization terms are used to manage mismatches between LoRA fine-tuning matrix pairs, we propose a specialized regularization term:

$$\left\| \mathbf{A}\mathbf{A}^\top - \frac{r}{m}\mathbf{B}^\top\mathbf{B} \right\|_{\mathrm{F}}^2 ,$$

where the factor $\frac{r}{m}$ compensates for the dimensionality mismatch and captures the inherent asymmetry between **A** and **B**. This regularization ensures that the norms of these components are appropriately balanced during fine-tuning, mitigating instability in forward propagation and enhancing the network's ability to represent complex features. Despite its effectiveness, iLoRA is a simple, plug-and-play method that requires only minor modifications and introduces negligible additional computational overhead. Moreover, we observe that standard gradient descent (GD) often fails to preserve the forward-pass imbalance between **A** and **B** in the backward pass, leading to inconsistencies in optimization dynamics. To resolve this, we apply the preconditioning method (Zhang and Pilanci, 2024), proving that under this approach, the backward pass maintains the same imbalance relationship as the forward pass, ensuring consistency throughout the optimization process.

We conduct extensive experiments with iLoRA, including fine-tuning GPT-2 on the E2E Natural Language Generation challenge (Novikova et al., 2017), Mistral 7B (Jiang et al., 2023) on the GLUE benchmark (Pilanci and Ergen, 2020), and diffusion models for image generation. Empirically, iLoRA demonstrates substantial performance improvements over traditional LoRA, with minimal additional computational cost. Furthermore, combining iLoRA with Riemannian Preconditioned LoRA, referred to as $\pi$LoRA, delivers significant performance gains across multiple tasks, showcasing the versatility of iLoRA. Additionally, our method exhibits enhanced robustness under varying learning rates, resulting in more stable and consistent training outcomes.

In summary, our contributions are as follows:

- We propose iLoRA, which introduces an imbalanced regularization strategy to capture the asymmetry between matrices **A** and **B** in LoRA, improving training stability and performance.

- We introduce $\pi$LoRA, which combines iLoRA with Riemannian preconditioning techniques (Zhang and Pilanci, 2024) to capture the inconsistency in backward propagation, ensuring alignment with the imbalance observed during the forward pass.

- Extensive experiments on GPT-2 (E2E), Mistral 7B (GLUE), and diffusion models demonstrate that both iLoRA and $\pi$LoRA achieve significant performance improvements over LoRA and preconditioned LoRA.

- Notably, our method incurs minimal computational overhead and requires only minor modifications to existing code, making it highly accessible and easy to adopt for a wide range of applications.

The rest of this paper is structured as follows: In Section 2, we give a comprehensive overview of related work. In Section 3, we provide the background and preliminaries. Section 4 provides the analysis of forward and backward propagation dynamics for LoRA, Section 5 details our iLoRA and $\pi$LoRA algorithms. Section 6 presents the experimental results. Finally, we summarize our contributions and discusses future work in Section 7.

## 2 RELATED WORK

**Low-Rank Adaptation:** In recent years, the rapid advancement of parameter-efficient fine-tuning (PEFT) techniques, particularly LoRA, has brought about numerous improvements. LoRA (Hu et al., 2022) is built on the principle that the updates required for fine-tuning large models can be represented as low-rank matrices, significantly reducing the number of trainable parameters. This foundational idea has sparked further innovations aimed at enhancing both the efficiency and effectiveness of LoRA-based methods. Several works (Hayou et al., 2024; Tian et al., 2024; Zhu et al., 2024) have focused on introducing asymmetry or refining rank allocation strategies to improve fine-tuning performance and efficiency. Moreover, preconditioning has been explored in works (Zhang and Pilanci, 2024) showing significant improvements in convergence and optimization. The gradient approximation technique (Wang et al., 2024) further enhances computational efficiency. Meanwhile, momentum filtering techniques (Chen et al., 2024) help mitigate catastrophic forgetting in large models.

In addition, various works have contributed novel architectures and optimizations, such as introducing multiple regeneration of fine-tuning matrices (Lialin et al., 2024; Xia et al., 2024; Zi et al., 2024), increasing parameter efficiency (Ren et al., 2024; Hao et al., 2024). These advancements solidify LoRA as a powerful and flexible tool for fine-tuning large-scale models.

**Preconditioning for Matrix Factorization:** Accelerating convergence via preconditioning has become a key approach in low-rank matrix factorization. The idea to precondition the gradient with $(\mathbf{A}\mathbf{A}^\top)^{-1}$ and $(\mathbf{B}^\top\mathbf{B})^{-1}$ was first suggested by Mishra et al. (2012), and later extended to Stochastic Gradient Descent (SGD) by Mishra and Sepulchre (2016). The convergence properties in the noiseless setting were studied by Li et al. (2018), leading to the development of the method now known as Scaled Gradient Descent (ScaledGD) for Matrix Factorization (Tong et al., 2021a). In subsequent work, Tong et al. (2021c) extended ScaledGD to subgradient methods, while Jia et al. (2023); Zhang et al. (2024; 2021) improved ScaledGD by using alternating optimization of $\mathbf{A}$ and $\mathbf{B}$, iterative hyperparameter updates and introduces time-varying preconditioning. These preconditioning methods, by addressing overparameterization and ill-conditioning, have become essential tools for improving the efficiency and accuracy of low-rank matrix estimation.

## 3 PRELIMINARIES

LoRA (Hu et al., 2022) builds on the insight that the updates needed for fine-tuning large pre-trained models are often approximately low-rank. Instead of updating the full-weight matrix during fine-tuning, LoRA offers a more efficient solution by decomposing the updates into the product of two low-rank matrices. This approach dramatically reduces the number of trainable parameters and lowers computational overhead, making it particularly well-suited for resource-intensive scenarios, such as fine-tuning Large Language Models (LLMs) and diffusion models.

In this paper, we use $\mathbf{W}$ to denote the weight matrix of a linear layer in the model. For example, in transformers, $\mathbf{W}$ can correspond to the $\mathbf{Q}$ (query), $\mathbf{K}$ (key), and $\mathbf{V}$ (value) matrices in the self-attention mechanism, or the weight matrices in the feedforward layers (MLP layers). LoRA's key idea is to express the fine-tuning update for each linear weight matrix as:

$$\mathbf{W}^\star = \mathbf{W}^{(0)} + \Delta\mathbf{W} = \mathbf{W}^{(0)} + \mathbf{B}\mathbf{A},$$

where $\mathbf{W}^\star, \mathbf{W}^{(0)} \in \mathbb{R}^{m \times n}$, $\mathbf{B} \in \mathbb{R}^{m \times r}$, and $\mathbf{A} \in \mathbb{R}^{r \times n}$, with $r \ll \min(m, n)$. The matrix $\mathbf{W}^{(0)}$ represents the pre-trained weights, which remain frozen during the fine-tuning process, while the low-rank matrices $\mathbf{A}$ and $\mathbf{B}$ are the newly introduced trainable parameters. This low-rank structure reduces memory and computation costs while enhancing the ability to efficiently adapt large pre-trained models to new tasks, achieving comparable accuracy to full-parameter fine-tuning.

## 4 ANALYSIS OF DYNAMICS AND IMBALANCED RELATIONSHIP IN LORA

Maintaining stable activations and gradients across layers is crucial to prevent issues such as vanishing or exploding gradients when training deep learning models (Glorot and Bengio, 2010). A key strategy to address this is ensuring that activation variances remain constant throughout the network (He

et al., 2015). In this section, we present key results for achieving this stability for both forward and backward propagation. First, we derive strategies for maintaining constant variance during forward propagation. To overcome the limitations of standard gradient descent in preserving proportional relationships between parameter updates during backward propagation, we integrate our method with a preconditioned approach. This combination improves training stability and convergence, addressing both forward and backward propagation challenges in LoRA training. For a more detailed analysis and proofs, please refer to Appendix B.

We now focus on a single matrix fine-tuning module, which serves as a crucial building block in the broader framework discussed earlier. Previously, we highlighted the importance of maintaining stable activations and gradients across multiple layers. Here, we extend these concepts to the LoRA-based architecture. Specifically, let $\mathbf{W}^{(0)} \in \mathbb{R}^{m \times n}$ represent the pre-trained weight matrix of a neural network layer, and let $\mathbf{A} \in \mathbb{R}^{r \times n}$ and $\mathbf{B} \in \mathbb{R}^{m \times r}$ be the low-rank matrices introduced during fine-tuning. Let $\mathbf{x} \in \mathbb{R}^n$ be an input vector. The forward propagation of the network can be expressed as:

$$\mathbf{f} = (\mathbf{W}^{(0)} + \mathbf{BA})\mathbf{x}. \tag{1}$$

The optimization objective is to minimize the loss:

$$L = \frac{1}{2}\|\mathbf{f}(\mathbf{x}) - \mathbf{y}\|^2, \tag{2}$$

where $\mathbf{y} \in \mathbb{R}^m$ is the target output vector. First, we start with the stability of forward propagation.

### 4.1 Variance Preservation in Forward Propagation

Define the intermediate activations as: $\mathbf{f}_1 = \mathbf{Ax}, \quad \mathbf{f}_2 = \mathbf{Bf}_1 = \mathbf{BAx}$. The variances of the elements in $\mathbf{f}_1$ and $\mathbf{f}_2$ remain in constant order and do not depend on the dimensions $n$, $m$, and $r$ (He et al., 2015) can ensure stable forward propagation in the network. If these variances were to scale with $n$, $m$, or $r$, it could lead to vanishing or exploding activations as the network depth or width increases, causing numerical instabilities and hindering effective training. Therefore, it is sufficient to control the variances of the elements of the parameter matrices $\mathbf{A}$ and $\mathbf{B}$ during training so that the variances of $\mathbf{f}_1$ and $\mathbf{f}_2$ remain constant order.

**Theorem 1** (Variance Preservation in Forward Propagation). *Let $\mathbf{x} \in \mathbb{R}^n$ be an input vector with i.i.d. elements of mean zero and variance $\sigma_x^2$. Under the assumptions that the elements of $\mathbf{A}$ and $\mathbf{B}$ have zero mean and variances $\sigma_A^2$ and $\sigma_B^2$ respectively, if the parameter variances satisfy:*

$$\sigma_A^2 = O\left(\frac{1}{n}\right), \quad \sigma_B^2 = O\left(\frac{1}{r}\right),$$

*the intermediate activations $\mathbf{f}_1$ and $\mathbf{f}_2$ have constant variances.*

**Remark 1:** Theorem 1 establishes the sufficient conditions for parameter variance to ensure that the variances of activations $\mathbf{f}_1$ and $\mathbf{f}_2$ remain constant order during forward propagation. This provides a clear objective for our regularization strategy, guiding us to maintain the stability of forward propagation by controlling the variances of the parameter matrices. Additionally, it is important to note that during stable forward propagation, the variances of the elements in the two fine-tuning matrices $\mathbf{A}$ and $\mathbf{B}$ are not identical. This variance asymmetry between the fine-tuning matrices $\mathbf{A}$ and $\mathbf{B}$ highlights a key characteristic of the fine-tuning process, indicating that $\mathbf{B}$ and $\mathbf{A}$ serve distinct roles in adapting the model, with $\mathbf{B}$ potentially requiring more variance than $\mathbf{A}$ to achieve balanced updates and maintain stability. This observation aligns with the findings of Hayou et al. (2024).

To control the imbalance between matrices $\mathbf{A}$ and $\mathbf{B}$, we propose modifying the commonly used balancing regularization term $\|\mathbf{A}\mathbf{A}^\top - \mathbf{B}^\top\mathbf{B}\|_F^2$ from the low-rank matrix factorization literature (Zhu et al., 2021). By imposing this regularization, we can restrict the degrees of freedom in matrix factorization, reducing the issue of infinitely many solutions due to the scalar associativity of matrix multiplication. While it might seem natural to directly consider the relationship between $\mathbf{A}$ and $\mathbf{B}$ or between $\mathbf{A}^\top\mathbf{A}$ and $\mathbf{B}^\top\mathbf{B}$. However, we instead focus on $\mathbf{A}\mathbf{A}^\top$ and $\mathbf{B}^\top\mathbf{B}$ because, although $\mathbf{A}$ and $\mathbf{B}$ have different dimensions, they share a common rank $r$. This common rank means that both $\mathbf{A}\mathbf{A}^\top$ and $\mathbf{B}^\top\mathbf{B}$ are square matrices of size $r \times r$. This dimensional consistency allows us to compare these terms and develop an effective regularization strategy.

Specifically, within the framework of Theorem 1, we introduce a scaling coefficient $\mu_1$ to control the balance between the matrices $\mathbf{A}$ and $\mathbf{B}$. We derive the results as follows:

**Corollary 1** (Scaling of $\mu_1$ with Matrix Dimensions). *Under the conditions of Theorem 1 , we have:*

$$\mathbb{E}[\mathbf{A}\mathbf{A}^\top] = \mu_1 \mathbb{E}[\mathbf{B}^\top\mathbf{B}], \quad \mu_1 = O\left(\frac{r}{m}\right).$$

**Remark 2:** Corollary 1 highlights that the proportionality constant $\mu_1$ scales with the ratio $\frac{r}{m}$, where $r$ is the rank of the matrix $\mathbf{A}$ and $m$ is the number of rows in $\mathbf{B}$. To compensate for this ratio, an imbalanced regularization term:

$$\left\|\mathbf{A}\mathbf{A}^\top - \frac{r}{m}\mathbf{B}^\top\mathbf{B}\right\|_{\mathrm{F}}^2,$$

is sufficient to maintain this imbalance relationship. This regularization ensures that the scaling of $\mathbf{A}$ and $\mathbf{B}$ is aligned according to their respective dimensions, preventing one matrix from dominating the update process and causing instability. By incorporating this term, we effectively manage the imbalance between $\mathbf{A}$ and $\mathbf{B}$.

## 4.2 LIMITATIONS OF STANDARD GRADIENT DESCENT

While the forward propagation ensures that activations have constant variance, it is equally important to maintain a stable update of the parameters during backpropagation. Specifically, we desire the changes in the parameter matrices to satisfy a similar proportional relationship: $\mathrm{d}(\mathbf{A}\mathbf{A}^\top) = \mu_2 \mathrm{d}(\mathbf{B}^\top\mathbf{B})$, where $\mathrm{d}(\cdot)$ denotes the infinitesimal change or differential in the matrix values during backpropagation. Our goal is to verify that $\mu_1$ (from forward propagation) and $\mu_2$ are of the same order, ensuring consistency between forward and backward propagation.

To explore this relationship further, we analyze how standard gradient descent affects the proportionality constant $\mu_2$ during backpropagation in Theorem 2.

**Theorem 2** (Proportional Inconsistency in Standard Gradient Descent). *Under the conditions of Theorem 1, for model Eq. (1), applying standard gradient descent to minimize loss Eq. (2) with a small learning rate $\eta$, the proportionality constant $\mu_2$ between $\mathrm{d}(\mathbf{A}\mathbf{A}^\top)$ and $\mathrm{d}(\mathbf{B}^\top\mathbf{B})$ satisfies:*

$$\mu_2 \approx 1,$$

*indicating an inherent balance in parameter updates due to the differing dimensions of $\mathbf{A}$ and $\mathbf{B}$.*

**Remark 3:** Theorem 2 demonstrates that, under standard gradient descent, there is a balance in parameter updates across $\mathrm{d}(\mathbf{A}\mathbf{A}^\top)$ and $\mathrm{d}(\mathbf{B}^\top\mathbf{B})$. However, this is inconsistent with the imbalance relationship observed during forward propagation for $\mathbf{A}\mathbf{A}^\top$ and $\mathbf{B}^\top\mathbf{B}$. As a result, $\mu_1$ and $\mu_2$ cannot be of the same order for gradient descent. This inconsistency suggests that standard gradient descent is insufficient for maintaining the desired proportional relationship between the updates of $\mathbf{A}$ and $\mathbf{B}$. Specifically, while forward propagation introduces an inherent imbalance between $\mathbf{A}$ and $\mathbf{B}$ due to their differing dimensions, backpropagation under standard gradient descent fails to account for this imbalance, leading to misaligned updates. To resolve this, we must scale gradient descent to ensure that the updates to $\mathbf{A}$ and $\mathbf{B}$ remain consistent with the proportionality introduced during forward propagation. This modification would allow us to align $\mu_2$ with $\mu_1$, ensuring that the imbalanced relationship between the matrices is maintained throughout the training process, ultimately leading to stable and effective updates.

## 4.3 SCALED GRADIENT DESCENT

To address the imbalance between parameters and updates identified in Theorem 2, we introduce a preconditioned gradient update method ScaledGD as proposed in Zhang and Pilanci (2024) which is also inspired by the previously discussed imbalanced relationship between $\mathbf{A}$ and $\mathbf{B}$. The scaled gradients are defined as:

$$\tilde{\nabla}_{\mathbf{A}} = (\mathbf{B}^\top\mathbf{B})^{-1}\frac{\partial L}{\partial \mathbf{A}}, \quad \tilde{\nabla}_{\mathbf{B}} = \frac{\partial L}{\partial \mathbf{B}}(\mathbf{A}\mathbf{A}^\top)^{-1}. \tag{3}$$

This modification leverages the inverses of the parameter covariance matrices to adjust the gradients, specifically aiming to resolve the imbalance between $\mathbf{A}$ and $\mathbf{B}$ and ensure that their updates remain proportional.

Using these scaled gradients, the parameter updates become:

$$\mathbf{A}_{\text{new}} = \mathbf{A} - \eta \tilde{\nabla}_{\mathbf{A}}, \quad \mathbf{B}_{\text{new}} = \mathbf{B} - \eta \tilde{\nabla}_{\mathbf{B}}. \tag{4}$$

Then we aim to verify whether the modified updates satisfy the relationship $\mathrm{d}(\mathbf{A}\mathbf{A}^\top) = \mu_2 \, \mathrm{d}(\mathbf{B}^\top\mathbf{B})$, and whether the proportionality constants $\mu_1$ from forward propagation and $\mu_2$ from backpropagation are of the same order. This verification ensures that the scaled gradient descent method maintains proportional consistency for stable parameter updates.

**Theorem 3** (Proportional Consistency in Scaled Gradient Descent). *Under the conditions of Theorem 1, for model Eq. (1), applying scaled gradient descent Eqs. (3) and (4) to minimize loss Eq. (2) with a small learning rate $\eta$, the proportionality constant $\mu_2$ between $\mathrm{d}(\mathbf{A}\mathbf{A}^\top)$ and $\mathrm{d}(\mathbf{B}^\top\mathbf{B})$ satisfies:*

$$\mu_2 \approx \mu_1 = O\left(\frac{r}{m}\right),$$

*ensuring that $\mu_1$ and $\mu_2$ are of the same order and thus maintaining consistency between forward and backward propagation.*

**Remark 4:** Theorem 3 confirms that the scaled gradient descent method effectively aligns the proportionality constants $\mu_2$, ensuring consistency in the proportion of parameters and updates between $\mathbf{A}$ and $\mathbf{B}$. This alignment addresses the inconsistency identified in standard gradient descent, promoting stable training by maintaining consistent proportional relationships during both forward and backward propagation.

## 5 IMBALANCE-REGULARIZED LoRA

In this section, we discuss how to incorporate the imbalanced regularization term derived in Section 4 into LoRA training by AdamW, forming the core of our iLoRA algorithm. Similar to the strategy of introducing weight decay in AdamW, the imbalanced regularization term we introduce only takes effect at the end of each iteration and does not interfere with the iteration of gradients and momentum. Specifically, consider the following regularization term scaled by a factor $\lambda$:

$$R(\mathbf{A}, \mathbf{B}) = \lambda \left\| \mathbf{A}\mathbf{A}^\top - \frac{r}{m}\mathbf{B}^\top\mathbf{B} \right\|_{\text{F}}^2.$$

The corresponding gradients are:

$$\nabla_{\mathbf{A}} R(\mathbf{A}, \mathbf{B}) = \lambda \left( \mathbf{A}\mathbf{A}^\top - \frac{r}{m}\mathbf{B}^\top\mathbf{B} \right) \mathbf{A},$$

$$\nabla_{\mathbf{B}} R(\mathbf{A}, \mathbf{B}) = \lambda \frac{r}{m}\mathbf{B} \left( \frac{r}{m}\mathbf{B}^\top\mathbf{B} - \mathbf{A}\mathbf{A}^\top \right).$$

After performing standard AdamW updates, we apply parameter update steps Eqs. (5) and (6) similar to weight decay. These gradients adjust the updates for $\mathbf{A}$ and $\mathbf{B}$ to ensure that the influence of imbalanced regularization is reflected in the parameter dynamics. The core steps of the algorithm are shown in Algorithm 1, while the complete procedure is provided in Algorithm 2 ( We use $\theta^A$ and $\theta^B$ to represent $\mathbf{A}$ and $\mathbf{B}$, respectively in the algorithm).

Our iLoRA algorithm ensures stability in forward propagation, but the inconsistency in backward propagation requires scaling the gradients to maintain proportionality. As shown in Section 4.3, by combining iLoRA with preconditioning methods, we introduce $\pi$**LoRA**, which leverages gradient scaling to ensure consistent parameter updates during both forward and backward propagation. Specifically, in $\pi$LoRA, we only need to replace line 3 in Algorithm 1 with preconditioning methods such as Scaled GD or Scaled AdamW in Zhang and Pilanci (2024)(see detail in Algorithm 3). This simple adjustment allows us to effectively combine the strengths of both iLoRA and preconditioning methods without altering their core structures, achieving the dual benefit of ensuring stability in forward propagation while resolving gradient inconsistencies in backward propagation. For other LoRA variants, incorporating the updates from Eqs. (5) and (6) after each iteration allows for a seamless combination of iLoRA with these variants, achieving a plug-and-play improvement in the algorithms.

---

**Algorithm 1** iLoRA: Imbalance-Regularized Low-Rank Adaptation

---

1: **Input:** $\eta$ (learning rate), $\lambda$ (regularization factor), $\theta_0$ (initial fine-tuning parameters), $r$ (rank), $m$ (pretrain matrix output dimension), $T$ (number of iterations).
2: **for** each iteration $t = 1, 2, \ldots, T$ **do**
3:     Perform standard AdamW updates for $\theta_{t-1}$: yielding $\theta_t^\star$
4:     # Or perform Scaled AdamW updates for $\theta_{t-1}$: yielding $\theta_t^\star$ in $\pi$LoRA
5:     Apply imbalanced regularization to $\theta_t^{A\star}$ and $\theta_t^{B\star}$:

$$\theta_t^A \leftarrow \theta_t^{A\star} - \eta \cdot \lambda \left( \theta_t^{A\star} \theta_t^{A\star\top} - \frac{r}{m} \theta_t^{B\star\top} \theta_t^{B\star} \right) \theta_t^{A\star} \tag{5}$$

$$\theta_t^B \leftarrow \theta_t^{B\star} - \eta \cdot \lambda \frac{r}{m} \theta_t^{B\star} \left( \frac{r}{m} \theta_t^{B\star\top} \theta_t^{B\star} - \theta_t^{A\star} \theta_t^{A\star\top} \right) \tag{6}$$

6: **end for**
7: **Output:** Optimized parameters $\theta_T$

---

# 6 EXPERIMENTS

In this section, we present a series of experiments to evaluate the performance and efficiency of our proposed methods. We start by fine-tuning large language models (LLMs) in Section 6.1, beginning with GPT-2 on the E2E dataset using iLoRA and $\pi$LoRA, followed by fine-tuning Mistral 7B[1] on the GLUE benchmark. Next, we showcase face generation results by fine-tuning a diffusion model using iLoRA and $\pi$LoRA in Section 6.2, demonstrating the application of our method beyond the language model. We then compare the training time of our method with standard LoRA, emphasizing that the additional computational overhead introduced by our method is negligible in Section 6.3. Finally, we conduct three ablation studies to explore the impact of key algorithmic details in Section 6.4. All experimental settings and additional experiments are provided in Appendix C.

## 6.1 LLM FINE-TUNING

In this section, we fine-tuned large language models using iLoRA and $\pi$LoRA methods. Specifically, we apply these methods to GPT-2 and Mistral 7B models across various tasks, datasets, LoRA ranks, and benchmarks. Empirically, we observe that the performance of iLoRA and $\pi$LoRA far exceed that of the baseline models, offering more than a $2\%$ improvement in performance. For details of our experiments, please refer to Appendix C.

### 6.1.1 GPT-2

In this section, we conducted fine-tuning experiments on the GPT-2 model using iLoRA and $\pi$LoRA. We followed the exact same experimental setup as Zhang and Pilanci (2024), ensuring consistency and comparability with previous methods. Detailed experimental settings and hyperparameters can be found in Appendix C.1.1. The results of fine-tuning GPT-2 with a LoRA rank of 4 on the E2E (Novikova et al., 2017) natural language generation challenge are summarized in Table 1. The table compares the performance of different methods, including the original LoRA, Preconditioned LoRA, and our proposed iLoRA and $\pi$LoRA methods, across five evaluation metrics: BLEU, NIST, METEOR (MET), ROUGE-L, and CIDEr. The results of LoRA and Preconditioned LoRA are referenced from Zhang and Pilanci (2024). From the table, we can see that iLoRA consistently improves over the original LoRA, demonstrating the effectiveness of our imbalanced regularization strategy. Moreover, $\pi$LoRA achieves the best performance across all evaluation metrics, surpassing both iLoRA and Preconditioned LoRA, demonstrating the benefits of combining imbalanced regularization with preconditioning. For additional experimental results and analyses, please refer to Appendix C.1.2.

---

[1]https://huggingface.co/mistralai/Mistral-7B-v0.1

Table 1: Results for LoRA fine-tuning of GPT-2 Model on the E2E Natural Language Generation Challenge with Different Methods. iLoRA and $\pi$LoRA outperform the original LoRA and Preconditioned LoRA across all evaluation metrics.

| Method | Rank | E2E | | | | |
|---|---|---|---|---|---|---|
| | | BLEU | NIST | MET | ROUGE-L | CIDEr |
| LoRA | 4 | 68.9 | 8.69 | 46.5 | 71.4 | 2.51 |
| iLoRA | 4 | 70.1 | 8.83 | **46.8** | 71.7 | 2.52 |
| Preconditioned LoRA | 4 | 69.6 | 8.77 | 46.6 | 71.8 | 2.52 |
| $\pi$LoRA | 4 | **70.8** | **8.89** | **46.8** | **72.1** | **2.54** |

### 6.1.2 MISTRAL 7B

In this section, we conducted fine-tuning experiments on the Mistral 7B model (Jiang et al., 2023) using iLoRA and $\pi$LoRA. Mistral 7B, released by the Mistral AI team, has demonstrated superior performance compared to Llama 2-13B on most benchmarks and has even surpassed Llama 1-34B on many tasks. As a result, it is considered one of the most powerful language models of its size to date. We followed the experimental setting from Zhang and Pilanci (2024) and applied our iLoRA and $\pi$LoRA methods to the General Language Understanding Evaluation (GLUE) benchmark (Wang, 2018). Detailed experimental settings and hyperparameters are provided in Appendix C.2.1.

The final results of fine-tuning Mistral 7B with a LoRA rank of 16 on the GLUE benchmark are shown in Table 2, with LoRA and Preconditioned LoRA results referenced from Zhang and Pilanci (2024). Our iLoRA method consistently outperforms the original LoRA across all tasks, with an average improvement of $1.44$ and $0.71$ over LoRA and Preconditioned LoRA, respectively. $\pi$LoRA delivers the best performance on nearly all tasks, achieving an average improvement of $2.82$ over LoRA and $2.09$ over Preconditioned LoRA. For further experimental results, please refer to Appendix C.2.

Table 2: Scores for LoRA fine-tuning of Mistral 7B Model on GLUE Benchmark with different methods. iLoRA and $\pi$LoRA show significant improvements respectively over LoRA and Preconditioned LoRA across all evaluation metrics.

| Method | Rank | GLUE | | | | | | | | | |
|---|---|---|---|---|---|---|---|---|---|---|---|
| | | MNLI | SST-2 | MRPC | CoLA | QNLI | QQP | RTE | STS-B | WNLI | Avg. |
| LoRA | 16 | 89.86 | 96.79 | 88.48 | 71.05 | 94.42 | 91.24 | 90.61 | 90.42 | 81.69 | 88.28 |
| iLoRA | 16 | 91.59 | 97.13 | 89.71 | 71.90 | 95.20 | 91.43 | 90.98 | 92.25 | 87.32 | 89.72 |
| Preconditioned LoRA | 16 | 90.68 | **97.25** | 89.46 | 71.30 | 94.67 | **92.22** | 91.34 | 91.10 | 83.10 | 89.01 |
| $\pi$LoRA | 16 | **91.61** | **97.25** | **90.44** | **71.97** | **95.37** | 91.44 | **91.70** | **92.35** | **88.73** | **91.10** |

## 6.2 DIFFUSION MODEL FINE-TUNING

Diffusion models are now widely applied in various image generation tasks, and LoRA has also been extensively used for fine-tuning these models. In this section, we conduct fine-tuning experiments on diffusion models to demonstrate the applicability of our methods (iLoRA and $\pi$LoRA) beyond large language models. Specifically, we experiment with the Mix-of-Show model (Gu et al., 2023), originally designed for multi-concept LoRA and proven to generate high-quality face images. To better visualize the differences between various LoRA optimization methods, we follow the settings from Zhang and Pilanci (2024) and disable embedding fine-tuning, focusing only on tuning the text encoders and U-Nets where LoRA factors are injected. We utilize 14 images of Potter from the original project repository, replacing the character name in the training images with "$\langle V_{\text{potter}} \rangle$". Fig. 1 presents the generation results for the prompt "a $\langle V_{\text{potter}} \rangle$ in front of eiffel tower" across different learning rates. Our methods (iLoRA and $\pi$LoRA) produce images that more accurately depict the prompt, and consistently perform well across different learning rates, demonstrating their effectiveness in generating higher-quality images and their robustness to changes in learning rates. The experimental settings are detailed in Appendix C.3.1. For additional results with different fusion coefficients and prompts, please refer to Appendix C.3.2.

| Learning Rate | LoRA | | iLoRA | | $\pi$LoRA | |
|---|---|---|---|---|---|---|
| | Image 1 | Image 2 | Image 1 | Image 2 | Image 1 | Image 2 |
| (5e-4, 5e-4) | | | | | | |
| (1e-4, 1e-4) | | | | | | |
| (5e-5, 5e-5) | | | | | | |

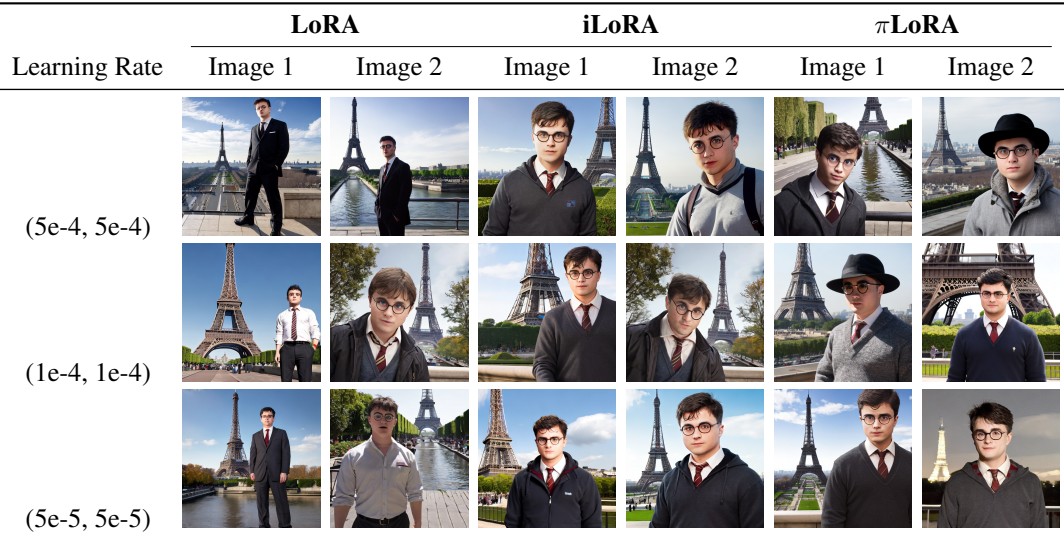

Figure 1: Comparison of images generated with LoRA, iLoRA, and $\pi$LoRA across different learning rates for the Mix-of-Show model. The three rows correspond to three different sets of learning rates for (text encoders, U-Nets): (5e-4, 5e-4), (1e-4, 1e-4) and (5e-5, 5e-5). The first and second columns show results from LoRA, the third and fourth columns show results from iLoRA, and the fifth and sixth columns show results from $\pi$LoRA. This layout demonstrates the robustness of each method under these learning rate settings.

## 6.3 RUNTIME COMPARISON

In this section, we investigate the impact of the additional computational cost introduced by imbalanced regularization in iLoRA and $\pi$LoRA algorithms. We perform fine-tuning of the GPT-2 model ($r = 4$) on the E2E NLG challenge and present a comparison of the training time between iLoRA, $\pi$LoRA, standard LoRA, and Preconditioned LoRA. Fig. 2 shows the runtime of the fine-tuning tasks using different algorithms on 1 * NVIDIA A100 GPU. As can be seen, the runtime differences among the four methods are minimal. This indicates that the regularization operations we introduced do not significantly increase the computational overhead, confirming the efficiency of our methods. Moreover, it is worth mentioning that the additional computational cost introduced by our regularization is smaller than the overhead introduced by the preconditioners in Zhang and Pilanci (2024).

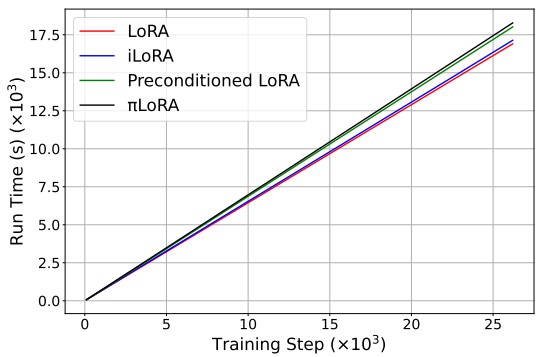

Figure 2: Runtime comparison of fine-tuning GPT-2 on E2E NLG challenge.

## 6.4 ABLATION STUDIES ON IMBALANCED COEFFICIENTS

To empirically verify the optimality of the imbalanced ratio $r/m$ in the iLoRA algorithm, we selected the CoLA task from the GLUE benchmark and conducted three ablation studies using the Mistral 7B model (a detailed introduction of the Mistral 7B model and GLUE benchmark are provide in Section 6.1.2). These studies evaluated the impact of different coefficients in the imbalanced regularization term and confirmed that the $r/m$ ratio in the iLoRA algorithm is indeed the most effective choice in practice.

In the first ablation study, we experimented with different multiplicative scaling factors $c$ applied to the ratio $r/m$, aiming to determine whether scaling this ratio could further enhance LoRA's

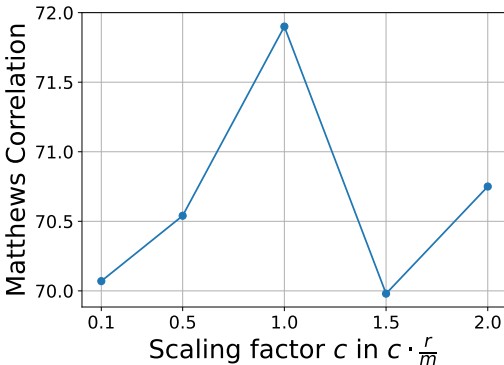 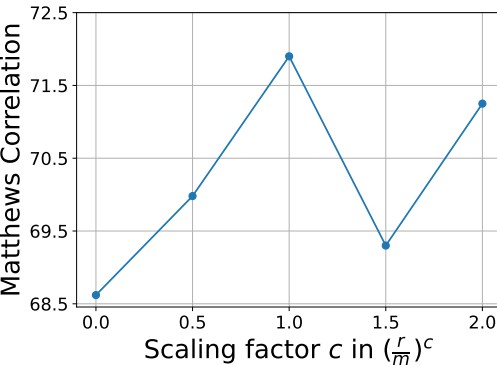

Figure 3: Ablation studies on the effects of multiplicative scaling factors and exponents applied to the ratio $r/m$ in LoRA. The left subplot shows the performance impact of different multiplicative scaling factors, while the right subplot illustrates the sensitivity of performance to varying exponents.

performance. The results, shown in the left subplot of Fig. 3, highlight the impact of various scaling factors on the model's performance. In the second ablation study, we varied the exponent $c$ applied to the ratio $r/m$ to investigate the sensitivity of performance to the exponentiation of the ratio. The outcomes of this experiment are displayed in the right subplot of Fig. 3.

The results from both ablation studies confirm that the ratio $r/m$ is empirically optimal. In the first study, performance peaks around $c = 0.9$ to $c = 1.0$, closely aligning with the ratio used in iLoRA. Similarly, in the second study, the exponent $c = 1$ achieves the highest performance, further validating that $r/m$ is the most effective ratio for imbalanced regularization in LoRA. These findings emphasize the significance of our theoretical analysis, demonstrating that the $r/m$ ratio not only has theoretical justification but also leads to superior empirical performance.

In the third ablation study, we investigated whether treating the imbalanced ratio coefficient $\zeta = \frac{r}{m}$ as a trainable parameter leads to better performance. Our findings reveal that the final learned value $\zeta_T$ is closely related to and slightly larger than the initial value $\zeta_0$. This is primarily due to the initialization of $B = 0$, which causes an imbalance in the regularization term early in training, leading to an increase in $\zeta$ at the beginning of the training progress. However, treating $\zeta$ as a trainable parameter produced slightly worse results compared to iLoRA. On the WNLI task, the performance for trainable $\zeta$ was 85.92, compared to 81.69 for LoRA and 87.32 for iLoRA. This result suggests that, while treating $\zeta$ as a trainable parameter relaxes certain constraints, it does not improve performance overall and is less effective than iLoRA.

## 7 CONCLUSION AND LIMITATIONS

In this paper, we proposed a plug-and-play fine-tuning method iLoRA(Imbalance-Regularized LoRA), which introduces an imbalanced regularization term to address the variance disparity between the fine-tuning matrices $\mathbf{A}$ and $\mathbf{B}$ in LoRA-based fine-tuning. This approach ensures that $\mathbf{A}\mathbf{A}^\top$ and $\mathbf{B}^\top\mathbf{B}$ maintain a proportional relationship, thereby enhancing stability in forward propagation. To address inconsistencies in backward propagation, we integrate iLoRA with preconditioning techniques to form $\pi$LoRA, utilizing gradient scaling to ensure consistent parameter updates in both forward and backward passes. Extensive experiments across various large-scale models and tasks demonstrate that iLoRA and $\pi$LoRA significantly improve training stability and model performance.

Nonetheless, certain limitations persist. While our methods address variance disparity within individual layers, they do not explicitly consider parameter imbalances across different layers, which may affect overall performance. Additionally, our approach focuses on balancing forward and backward propagation but does not account for optimization dynamics like momentum in adaptive optimizers. We also aim to investigate the effectiveness of our methods across a wider range of architectures, such as large vision-language models. Addressing these challenges will be the focus of future work.

## ETHICS STATEMENT

This work focuses on improving fine-tuning methods for large language models (LLMs) using iLoRA and $\pi$LoRA. Our research does not involve any direct human subjects or the collection of sensitive data. However, we acknowledge the potential risks associated with deploying fine-tuned LLMs, such as unintended biases, harmful outputs, and privacy concerns. These risks are primarily related to how LLMs are used post-deployment and the quality of datasets used during fine-tuning. We have followed best practices to mitigate such risks by using publicly available datasets (E2E, GLUE) and ensuring the results focus on model performance. The methods proposed in this paper are intended to contribute to more efficient and stable fine-tuning but should be used with caution in applications that could lead to ethical concerns, particularly when deployed in sensitive environments. No conflicts of interest or external sponsorship influenced this research.

## REPRODUCIBILITY STATEMENT

To ensure the reproducibility of our work, we provide detailed experimental settings and hyperparameters in Appendix C.1.1 and Appendix C.2.1. The source code for iLoRA and $\pi$LoRA will be made publicly available in the camera-ready version. For theoretical results, we provide a complete analysis and proof in Appendix B. All datasets used in our experiments (E2E, GLUE) are publicly available, and the specific data processing steps for each experiment are detailed in Appendix C. We have made every effort to ensure that all components of our work are reproducible and transparent.

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

## A  FULL VERSION OF THE ALGORITHM

---

**Algorithm 2** iLoRA: Imbalance-Regularized Low-Rank Adaptation

---

1: **Input:** $\eta$ (learning rate), $\beta_1, \beta_2 \in [0, 1)$ (exponential decay rates for moment estimates), $\lambda$ (regularization factor), $\lambda^\star$ (weight decay factor), $\epsilon$ (small constant for numerical stability), $\theta_0$ (initial fine-tuning parameters), $L(\theta)$ (objective function), $r$ (rank), $m$ (pretrain matrix output dimension), $T$ (number of iterations).

2: **Initialize:** $m_0 \leftarrow 0$ (initial first moment), $v_0 \leftarrow 0$ (initial second moment), $t \leftarrow 0$ (initial timestep)

3: **for** each iteration $t = 1, 2, \ldots, T$ **do**

4:    Compute gradient: $g_t \leftarrow \nabla_\theta L(\theta_{t-1})$

5:    Update biased first moment estimate: $m_t \leftarrow \beta_1 m_{t-1} + (1 - \beta_1) g_t$

6:    Update biased second moment estimate: $v_t \leftarrow \beta_2 v_{t-1} + (1 - \beta_2) g_t^2$

7:    Compute bias-corrected first moment estimate: $\hat{m}_t \leftarrow \frac{m_t}{1-\beta_1^t}$

8:    Compute bias-corrected second moment estimate: $\hat{v}_t \leftarrow \frac{v_t}{1-\beta_2^t}$

9:    Perform AdamW update: $\theta_t^\star \leftarrow \theta_{t-1} - \eta \left( \frac{\hat{m}_t}{\sqrt{\hat{v}_t}+\epsilon} + \lambda^\star \theta_{t-1} \right)$

10:    Apply imbalanced regularization to $\theta_t^{A\star}$:

$$\theta_t^A \leftarrow \theta_t^{A\star} - \eta \cdot \lambda \left( \theta_t^{A\star} \theta_t^{A\star\top} - \frac{r}{m} \theta_t^{B\star\top} \theta_t^{B\star} \right) \theta_t^{A\star}$$

11:    Apply imbalanced regularization to $\theta_t^{B\star}$:

$$\theta_t^B \leftarrow \theta_t^{B\star} - \eta \cdot \lambda \frac{r}{m} \theta_t^{B\star} \left( \frac{r}{m} \theta_t^{B\star\top} \theta_t^{B\star} - \theta_t^{A\star} \theta_t^{A\star\top} \right)$$

12: **end for**

13: **Output:** Optimized parameters $\theta_T$

---

## B  PROOFS AND DETAILED ANALYSIS

### B.1  PROOF OF THEOREM 1

*Proof.* Starting with the intermediate variable $\mathbf{f}_1 = \mathbf{A}\mathbf{x}$, under the assumption that $\mathbf{x}$ has i.i.d. elements with mean zero and variance $\sigma_x^2$, and that the elements $A_{kj}$ of matrix $\mathbf{A}$ are i.i.d. with mean zero and variance $\sigma_A^2$, the variance of each element $f_{1k}$ of $\mathbf{f}_1$ is computed as follows:

$$f_{1k} = \sum_{j=1}^n A_{kj} x_j.$$

Since $A_{kj}$ and $x_j$ are independent and both have zero mean, the variance of $f_{1k}$ is:

$$\text{Var}(f_{1k}) = \sum_{j=1}^n \text{Var}(A_{kj} x_j) = \sum_{j=1}^n \text{Var}(A_{kj}) \cdot \text{Var}(x_j) = n\sigma_A^2 \sigma_x^2.$$

To maintain a constant variance for $f_{1k}$ that is independent of the dimension $n$, it is necessary that:

$$\sigma_A^2 = O\left(\frac{1}{n}\right).$$

Next, consider the intermediate variable $\mathbf{f}_2 = \mathbf{B}\mathbf{f}_1$, where the elements $B_{ik}$ of matrix $\mathbf{B}$ are i.i.d. with mean zero and variance $\sigma_B^2$. The variance of each element $f_{2i}$ of $\mathbf{f}_2$ is:

$$f_{2i} = \sum_{k=1}^r B_{ik} f_{1k}.$$

---

**Algorithm 3** $\pi$LoRA: Preconditioned Imbalance-Regularized Low-Rank Adaptation

---

1: **Input:** $\eta$ (learning rate), $\beta_1, \beta_2 \in [0, 1)$ (exponential decay rates for moment estimates), $\lambda$ (regularization factor), $\lambda^\star$ (weight decay factor), $\epsilon$ (small constant for numerical stability), $\theta_0$ (initial fine-tuning parameters), $L(\theta)$ (objective function), $r$ (rank), $m$ (pretrain matrix output dimension), $T$ (number of iterations).

2: **Initialize:** $m_0 \leftarrow 0$ (initial first moment), $v_0 \leftarrow 0$ (initial second moment), $t \leftarrow 0$ (initial timestep)

3: **for** each iteration $t = 1, 2, \ldots, T$ **do**

4:     Compute gradient: $g_t \leftarrow \nabla_\theta L(\theta_{t-1})$

5:     Scale the gradient:
$$\tilde{g}_t^A \leftarrow (\theta_{t-1}^{B\top}\theta_{t-1}^B)^{-1}g_t^A$$
$$\tilde{g}_t^B \leftarrow g_t^B(\theta_{t-1}^A\theta_{t-1}^{A\top})^{-1}$$

6:     Update biased first moment estimate: $m_t \leftarrow \beta_1 m_{t-1} + (1-\beta_1)\tilde{g}_t$

7:     Update biased second moment estimate: $v_t \leftarrow \beta_2 v_{t-1} + (1-\beta_2)\tilde{g}_t^2$

8:     Compute bias-corrected first moment estimate: $\hat{m}_t \leftarrow \frac{m_t}{1-\beta_1^t}$

9:     Compute bias-corrected second moment estimate: $\hat{v}_t \leftarrow \frac{v_t}{1-\beta_2^t}$

10:     Perform AdamW update: $\theta_t^\star \leftarrow \theta_{t-1} - \eta\left(\frac{\hat{m}_t}{\sqrt{\hat{v}_t}+\epsilon} + \lambda^\star\theta_{t-1}\right)$

11:     Apply imbalanced regularization to $\theta_t^{A\star}$:
$$\theta_t^A \leftarrow \theta_t^{A\star} - \eta \cdot \lambda\left(\theta_t^{A\star}\theta_t^{A\star\top} - \frac{r}{m}\theta_t^{B\star\top}\theta_t^{B\star}\right)\theta_t^{A\star}$$

12:     Apply imbalanced regularization to $\theta_t^{B\star}$:
$$\theta_t^B \leftarrow \theta_t^{B\star} - \eta \cdot \lambda\frac{r}{m}\theta_t^{B\star}\left(\frac{r}{m}\theta_t^{B\star\top}\theta_t^{B\star} - \theta_t^{A\star}\theta_t^{A\star\top}\right)$$

13: **end for**

14: **Output:** Optimized parameters $\theta_T$

---

Since $B_{ik}$ and $f_{1k}$ are independent, and $\mathrm{Var}(f_{1k}) = n\sigma_A^2\sigma_x^2$, the variance of $f_{2i}$ is:

$$\mathrm{Var}(f_{2i}) = \sum_{k=1}^{r} \mathrm{Var}(B_{ik}f_{1k}) = \sum_{k=1}^{r} \mathrm{Var}(B_{ik}) \cdot \mathrm{Var}(f_{1k}) = r\sigma_B^2 n\sigma_A^2\sigma_x^2.$$

Substituting $\sigma_A^2 = O\left(\frac{1}{n}\right)$ into the equation:

$$\mathrm{Var}(f_{2i}) = r\sigma_B^2 n\left(O\left(\frac{1}{n}\right)\right)\sigma_x^2 = r\sigma_B^2\sigma_x^2.$$

To maintain a constant variance for $f_{2i}$ that is independent of the dimension $r$, it is necessary that:

$$\sigma_B^2 = O\left(\frac{1}{r}\right).$$

This completes the proof of Theorem 1. $\qquad\square$

### B.2 PROOF OF COROLLARY 1

*Proof.* Under the conditions of Theorem 1, we have established that:

$$\sigma_A^2 = O\left(\frac{1}{n}\right), \quad \sigma_B^2 = O\left(\frac{1}{r}\right).$$

First, compute the expected values of $\mathbf{A}\mathbf{A}^\top$ and $\mathbf{B}^\top\mathbf{B}$:

$$\mathbb{E}[\mathbf{A}\mathbf{A}^\top] = n\sigma_A^2\mathbf{I}_r = O(1)\mathbf{I}_r,$$

$$\mathbb{E}[\mathbf{B}^\top\mathbf{B}] = m\sigma_B^2\mathbf{I}_r = O\left(\frac{m}{r}\right)\mathbf{I}_r,$$

where $\mathbf{I}_r$ is the $r \times r$ identity matrix.

This gives:

$$\mu_1 = O\left(\frac{r}{m}\right).$$

Therefore, the proportionality constant $\mu_1$ scales as $O\left(\frac{r}{m}\right)$, reflecting the relationship between the dimensions of matrices $\mathbf{A}$ and $\mathbf{B}$. $\qquad\square$

### B.3 PROOF OF THEOREM 2

*Proof.* First, we use $\mathbf{e} = \mathbf{f}(\mathbf{x}) - \mathbf{y}$ to denote the error vector and rewrite the mean squared loss as the following:

$$L = \frac{1}{2}\|\mathbf{f}(\mathbf{x}) - \mathbf{y}\|^2 = \frac{1}{2}\mathbf{e}^\top\mathbf{e},$$

Next, we compute the gradients with respect to $\mathbf{A}$ and $\mathbf{B}$:

$$\frac{\partial L}{\partial \mathbf{A}} = \mathbf{B}^\top\mathbf{e}\mathbf{x}^\top, \qquad \frac{\partial L}{\partial \mathbf{B}} = \mathbf{e}(\mathbf{A}\mathbf{x})^\top. \tag{7}$$

By standard gradient descent with learning rate $\eta$:

$$\mathbf{A}_{\mathrm{new}} = \mathbf{A} - \eta\frac{\partial L}{\partial \mathbf{A}}, \quad \mathbf{B}_{\mathrm{new}} = \mathbf{B} - \eta\frac{\partial L}{\partial \mathbf{B}}.$$

Next, we compute the change in $\mathbf{A}\mathbf{A}^\top$:

$$\mathrm{d}(\mathbf{A}\mathbf{A}^\top) = \mathbf{A}_{\mathrm{new}}\mathbf{A}_{\mathrm{new}}^\top - \mathbf{A}\mathbf{A}^\top$$

$$= (\mathbf{A} - \eta\frac{\partial L}{\partial \mathbf{A}})(\mathbf{A} - \eta\frac{\partial L}{\partial \mathbf{A}})^\top - \mathbf{A}\mathbf{A}^\top$$

$$\approx -\eta\left(\mathbf{A}\left(\frac{\partial L}{\partial \mathbf{A}}\right)^\top + \left(\frac{\partial L}{\partial \mathbf{A}}\right)\mathbf{A}^\top\right),$$

where we neglect the $\eta^2$ term as $\eta$ is small. Similarly, compute the change in $\mathbf{B}^\top \mathbf{B}$ and we get:

$$\mathrm{d}(\mathbf{B}^\top \mathbf{B}) \approx -\eta \left( \mathbf{B}^\top \left( \frac{\partial L}{\partial \mathbf{B}} \right) + \left( \frac{\partial L}{\partial \mathbf{B}} \right)^\top \mathbf{B} \right).$$

Therefore, to satisfy the desired relationship $\mathrm{d}(\mathbf{A}\mathbf{A}^\top) = \mu_2 \, \mathrm{d}(\mathbf{B}^\top \mathbf{B})$, we need:

$$\mathbf{A} \left( \frac{\partial L}{\partial \mathbf{A}} \right)^\top + \left( \frac{\partial L}{\partial \mathbf{A}} \right) \mathbf{A}^\top = \mu_2 \left( \mathbf{B}^\top \left( \frac{\partial L}{\partial \mathbf{B}} \right) + \left( \frac{\partial L}{\partial \mathbf{B}} \right)^\top \mathbf{B} \right). \tag{8}$$

Substituting the expressions for the gradients from Eq. (7) to the left-hand side of Eq. (8), we get:

$$\mathbf{A} \left( \frac{\partial L}{\partial \mathbf{A}} \right)^\top + \left( \frac{\partial L}{\partial \mathbf{A}} \right) \mathbf{A}^\top = \mathbf{A} \left( \mathbf{B}^\top \mathbf{e}\mathbf{x}^\top \right)^\top + \left( \mathbf{B}^\top \mathbf{e}\mathbf{x}^\top \right) \mathbf{A}^\top$$

$$= \mathbf{A} \left( \mathbf{x}\mathbf{e}^\top \mathbf{B} \right) + \mathbf{B}^\top \mathbf{e}\mathbf{x}^\top \mathbf{A}^\top$$

$$= \mathbf{A}\mathbf{x}\mathbf{e}^\top \mathbf{B} + \mathbf{B}^\top \mathbf{e}\mathbf{x}^\top \mathbf{A}^\top.$$

Similarly, the right-hand side of Eq. (8) becomes:

$$\mu_2 \left( \mathbf{B}^\top \frac{\partial L}{\partial \mathbf{B}} + \left( \frac{\partial L}{\partial \mathbf{B}} \right)^\top \mathbf{B} \right) = \mu_2 \left( \mathbf{B}^\top \mathbf{e}(\mathbf{A}\mathbf{x})^\top + \left( \mathbf{e}(\mathbf{A}\mathbf{x})^\top \right)^\top \mathbf{B} \right)$$

$$= \mu_2 \left( \mathbf{B}^\top \mathbf{e}\mathbf{x}^\top \mathbf{A}^\top + \mathbf{A}\mathbf{x}\mathbf{e}^\top \mathbf{B} \right).$$

Therefore, Eq. (8) becomes:

$$\mathbf{A}\mathbf{x}\mathbf{e}^\top \mathbf{B} + \mathbf{B}^\top \mathbf{e}\mathbf{x}^\top \mathbf{A}^\top \approx \mu_2 \left( \mathbf{B}^\top \mathbf{e}\mathbf{x}^\top \mathbf{A}^\top + \mathbf{A}\mathbf{x}\mathbf{e}^\top \mathbf{B} \right).$$

By rearranging the term, we have:

$$(1 - \mu_2) \left( \mathbf{A}\mathbf{x}\mathbf{e}^\top \mathbf{B} + \mathbf{B}^\top \mathbf{e}\mathbf{x}^\top \mathbf{A}^\top \right) \approx 0.$$

Therefore, unless $\mu_2 \approx 1$, this equality does not generally hold with standard gradient descent.

$\square$

### B.4 Proof of Theorem 3

*Proof.* We aim to verify that the scaled gradient updates satisfy $\mathrm{d}(\mathbf{A}\mathbf{A}^\top) = \mu_2 \, \mathrm{d}(\mathbf{B}^\top \mathbf{B})$, and that $\mu_1$ and $\mu_2$ are of the same order.

The scaled gradients are:

$$\tilde{\nabla}_\mathbf{A} = (\mathbf{B}^\top \mathbf{B})^{-1} \frac{\partial L}{\partial \mathbf{A}}, \quad \tilde{\nabla}_\mathbf{B} = \frac{\partial L}{\partial \mathbf{B}} (\mathbf{A}\mathbf{A}^\top)^{-1}.$$

The parameter updates are:

$$\mathbf{A}_\text{new} = \mathbf{A} - \eta \tilde{\nabla}_\mathbf{A}, \quad \mathbf{B}_\text{new} = \mathbf{B} - \eta \tilde{\nabla}_\mathbf{B}.$$

Then, we compute the change in $\mathbf{A}\mathbf{A}^\top$:

$$\mathrm{d}(\mathbf{A}\mathbf{A}^\top) = \mathbf{A}_\text{new} \mathbf{A}_\text{new}^\top - \mathbf{A}\mathbf{A}^\top$$

$$= (\mathbf{A} - \eta \tilde{\nabla}_\mathbf{A})(\mathbf{A} - \eta \tilde{\nabla}_\mathbf{A})^\top - \mathbf{A}\mathbf{A}^\top$$

$$\approx -\eta \left( \mathbf{A}\tilde{\nabla}_\mathbf{A}^\top + \tilde{\nabla}_\mathbf{A} \mathbf{A}^\top \right), \quad \text{(neglecting } \eta^2 \text{ terms)}$$

$$= -\eta \left( \mathbf{A} \left( \frac{\partial L}{\partial \mathbf{A}} \right)^\top (\mathbf{B}^\top \mathbf{B})^{-1} + (\mathbf{B}^\top \mathbf{B})^{-1} \frac{\partial L}{\partial \mathbf{A}} \mathbf{A}^\top \right).$$

Similarly, compute the change in $\mathbf{B}^\top \mathbf{B}$:

$$
\begin{aligned}
\mathrm{d}(\mathbf{B}^\top \mathbf{B}) &= \mathbf{B}_{\text{new}}^\top \mathbf{B}_{\text{new}} - \mathbf{B}^\top \mathbf{B} \\
&= (\mathbf{B} - \eta \tilde{\nabla}_{\mathbf{B}})^\top (\mathbf{B} - \eta \tilde{\nabla}_{\mathbf{B}}) - \mathbf{B}^\top \mathbf{B} \\
&\approx -\eta \left( \mathbf{B}^\top \tilde{\nabla}_{\mathbf{B}} + \tilde{\nabla}_{\mathbf{B}}^\top \mathbf{B} \right), \quad \text{(neglecting } \eta^2 \text{ terms)} \\
&= -\eta \left( \mathbf{B}^\top \frac{\partial L}{\partial \mathbf{B}} (\mathbf{A}\mathbf{A}^\top)^{-1} + (\mathbf{A}\mathbf{A}^\top)^{-1} \left( \frac{\partial L}{\partial \mathbf{B}} \right)^\top \mathbf{B} \right).
\end{aligned}
$$

By substituting the partial derivatives in Eq. (7) into $\mathrm{d}(\mathbf{A}\mathbf{A}^\top)$, we get:

$$
\begin{aligned}
\mathrm{d}(\mathbf{A}\mathbf{A}^\top) &= -\eta \left( \mathbf{A} \left( \frac{\partial L}{\partial \mathbf{A}} \right)^\top (\mathbf{B}^\top \mathbf{B})^{-1} + (\mathbf{B}^\top \mathbf{B})^{-1} \frac{\partial L}{\partial \mathbf{A}} \mathbf{A}^\top \right) \\
&= -\eta \left( \mathbf{A} \left( \mathbf{B}^\top \mathbf{e}\mathbf{x}^\top \right)^\top (\mathbf{B}^\top \mathbf{B})^{-1} + (\mathbf{B}^\top \mathbf{B})^{-1} \mathbf{B}^\top \mathbf{e}\mathbf{x}^\top \mathbf{A}^\top \right) \\
&= -\eta \left( \mathbf{A}\mathbf{x}\mathbf{e}^\top \mathbf{B}(\mathbf{B}^\top \mathbf{B})^{-1} + (\mathbf{B}^\top \mathbf{B})^{-1} \mathbf{B}^\top \mathbf{e}\mathbf{x}^\top \mathbf{A}^\top \right).
\end{aligned}
$$

Similarly, substituting into $\mathrm{d}(\mathbf{B}^\top \mathbf{B})$:

$$
\begin{aligned}
\mathrm{d}(\mathbf{B}^\top \mathbf{B}) &= -\eta \left( \mathbf{B}^\top \left( \mathbf{e}(\mathbf{A}\mathbf{x})^\top \right) (\mathbf{A}\mathbf{A}^\top)^{-1} + (\mathbf{A}\mathbf{A}^\top)^{-1} \left( \mathbf{e}(\mathbf{A}\mathbf{x})^\top \right)^\top \mathbf{B} \right) \\
&= -\eta \left( \mathbf{B}^\top \mathbf{e}\mathbf{x}^\top \mathbf{A}^\top (\mathbf{A}\mathbf{A}^\top)^{-1} + (\mathbf{A}\mathbf{A}^\top)^{-1} \mathbf{A}\mathbf{x}\mathbf{e}^\top \mathbf{B} \right).
\end{aligned}
$$

Next, we perform Singular Value Decomposition (SVD) on matrices $\mathbf{A}$ and $\mathbf{B}$, (the dimension of $\mathbf{\Sigma}_A$ and $\mathbf{\Sigma}_B$ are $r \times r$, the elements are arranged from large to small, and it is assumed that there is no multiplicity):

$$
\mathbf{A} = \mathbf{U}_A \mathbf{\Sigma}_A \mathbf{V}_A^\top, \quad \mathbf{B} = \mathbf{U}_B \mathbf{\Sigma}_B \mathbf{V}_B^\top.
$$

Given the matrix relationship $\mathbf{A}\mathbf{A}^\top = \mu_1 \mathbf{B}^\top \mathbf{B}$, we compute $\mathbf{A}\mathbf{A}^\top$ and $\mathbf{B}^\top \mathbf{B}$ as follows:

$$
\begin{aligned}
\mathbf{A}\mathbf{A}^\top &= (\mathbf{U}_A \mathbf{\Sigma}_A \mathbf{V}_A^\top)(\mathbf{V}_A \mathbf{\Sigma}_A^\top \mathbf{U}_A^\top) = \mathbf{U}_A \mathbf{\Sigma}_A \mathbf{\Sigma}_A^\top \mathbf{U}_A^\top, \\
\mathbf{B}^\top \mathbf{B} &= (\mathbf{V}_B \mathbf{\Sigma}_B^\top \mathbf{U}_B^\top)(\mathbf{U}_B \mathbf{\Sigma}_B \mathbf{V}_B^\top) = \mathbf{V}_B \mathbf{\Sigma}_B^\top \mathbf{\Sigma}_B \mathbf{V}_B^\top.
\end{aligned}
$$

Substitute into the matrix relationship:

$$
\mathbf{U}_A \mathbf{\Sigma}_A \mathbf{\Sigma}_A^\top \mathbf{U}_A^\top = \mu_1 \mathbf{V}_B \mathbf{\Sigma}_B^\top \mathbf{\Sigma}_B \mathbf{V}_B^\top
$$

Multiply both sides on the left by $\mathbf{U}_A^\top$ and on the right by $\mathbf{U}_A$:

$$
\begin{aligned}
\mathbf{U}_A^\top (\mathbf{U}_A \mathbf{\Sigma}_A \mathbf{\Sigma}_A^\top \mathbf{U}_A^\top) \mathbf{U}_A &= \mu_1 \mathbf{U}_A^\top (\mathbf{V}_B \mathbf{\Sigma}_B^\top \mathbf{\Sigma}_B \mathbf{V}_B^\top) \mathbf{U}_A \\
\mathbf{\Sigma}_A \mathbf{\Sigma}_A^\top &= \mu_1 (\mathbf{U}_A^\top \mathbf{V}_B) \mathbf{\Sigma}_B^\top \mathbf{\Sigma}_B (\mathbf{V}_B^\top \mathbf{U}_A)
\end{aligned}
$$

Since $\mathbf{U}_A^\top \mathbf{U}_A = \mathbf{I}$, we have $\mathbf{U}_A^\top \mathbf{V}_B = \mathbf{Q}$, where $\mathbf{Q}$ is an orthogonal matrix. Because $\mathbf{\Sigma}_A \mathbf{\Sigma}_A^\top$ and $\mathbf{\Sigma}_B^\top \mathbf{\Sigma}_B$ are diagonal, we require that:

$$
\mathbf{\Sigma}_A \mathbf{\Sigma}_A^\top = \mu_1 \mathbf{Q} \mathbf{\Sigma}_B^\top \mathbf{\Sigma}_B \mathbf{Q}^\top.
$$

For the equality of diagonal matrices, we must have $\mathbf{Q} \pm \mathbf{I}$. Without loss of generality, we consider $\mathbf{Q} = \mathbf{I}$, which implies $\mathbf{U}_A = \mathbf{V}_B$.

Thus, we have the alignment of singular vectors:

$$
\mathbf{U}_A = \mathbf{V}_B.
$$

Also, the proportionality of singular values:

$$
\mathbf{\Sigma}_A \mathbf{\Sigma}_A^\top = \mu_1 \mathbf{\Sigma}_B^\top \mathbf{\Sigma}_B
$$

$$(\boldsymbol{\Sigma}_A \boldsymbol{\Sigma}_A^\top)_{ii} = \mu_1 (\boldsymbol{\Sigma}_B^\top \boldsymbol{\Sigma}_B)_{ii}$$
$$\sigma_{A,i}^2 = \mu_1 \sigma_{B,i}^2$$
$$\sigma_{A,i} = \sqrt{\mu_1}\,\sigma_{B,i}$$

Therefore, the singular values satisfy:

$$\boldsymbol{\Sigma}_A = \sqrt{\mu_1}\,\boldsymbol{\Sigma}_B\,.$$

Substituting the SVD Decomposition of $\mathbf{A}$ and $\mathbf{B}$ into $(\mathbf{A}\mathbf{A}^\top)^{-1}$ and $(\mathbf{B}^\top\mathbf{B})^{-1}$:

$$(\mathbf{A}\mathbf{A}^\top)^{-1} = \mathbf{U}_A (\boldsymbol{\Sigma}_A \boldsymbol{\Sigma}_A^\top)^{-1} \mathbf{U}_A^\top,$$
$$(\mathbf{B}^\top\mathbf{B})^{-1} = \mathbf{V}_B (\boldsymbol{\Sigma}_B^\top \boldsymbol{\Sigma}_B)^{-1} \mathbf{V}_B^\top.$$

We have that both $\boldsymbol{\Sigma}_A$ and $\boldsymbol{\Sigma}_B$ are diagonal and full-rank. And $\mathbf{U}_A^\top \mathbf{U}_A = \mathbf{I}$ and $\mathbf{V}_B^\top \mathbf{V}_B = \mathbf{I}$, this allows us to further simplify the expressions for $\mathrm{d}(\mathbf{A}\mathbf{A}^\top)$ and $\mathrm{d}(\mathbf{B}^\top\mathbf{B})$.

Starting with $\mathrm{d}(\mathbf{A}\mathbf{A}^\top)$:

$$\begin{aligned}
\mathrm{d}(\mathbf{A}\mathbf{A}^\top) = &- \eta(\mathbf{U}_A \boldsymbol{\Sigma}_A \mathbf{V}_A^\top \mathbf{x} \mathbf{e}^\top \mathbf{U}_B \boldsymbol{\Sigma}_B \mathbf{V}_B^\top \mathbf{V}_B (\boldsymbol{\Sigma}_B^\top \boldsymbol{\Sigma}_B)^{-1} \mathbf{V}_B^\top \\
&+ \mathbf{V}_B (\boldsymbol{\Sigma}_B^\top \boldsymbol{\Sigma}_B)^{-1} \mathbf{V}_B^\top \mathbf{V}_B \boldsymbol{\Sigma}_B^\top \mathbf{U}_B^\top \mathbf{e} \mathbf{x}^\top \mathbf{V}_A \boldsymbol{\Sigma}_A^\top \mathbf{U}_A^\top) \\
= &- \eta \left( \mathbf{U}_A \boldsymbol{\Sigma}_A \mathbf{V}_A^\top \mathbf{x} \mathbf{e}^\top \mathbf{U}_B \boldsymbol{\Sigma}_B^{-1} \mathbf{V}_B^\top + \mathbf{V}_B \boldsymbol{\Sigma}_B^{-1} \mathbf{U}_B^\top \mathbf{e} \mathbf{x}^\top \mathbf{V}_A \boldsymbol{\Sigma}_A^\top \mathbf{U}_A^\top \right).
\end{aligned}$$

Similarly, for $\mathrm{d}(\mathbf{B}^\top\mathbf{B})$:

$$\begin{aligned}
\mathrm{d}(\mathbf{B}^\top\mathbf{B}) = &- \eta(\mathbf{V}_B \boldsymbol{\Sigma}_B \mathbf{U}_B^\top \mathbf{e} \mathbf{x}^\top \mathbf{V}_A \boldsymbol{\Sigma}_A \mathbf{U}_A^\top \mathbf{U}_A (\boldsymbol{\Sigma}_A \boldsymbol{\Sigma}_A^\top)^{-1} \mathbf{U}_A^\top \\
&+ \mathbf{U}_A (\boldsymbol{\Sigma}_A \boldsymbol{\Sigma}_A^\top)^{-1} \mathbf{U}_A^\top \mathbf{U}_A \boldsymbol{\Sigma}_A \mathbf{V}_A^\top \mathbf{x} \mathbf{e}^\top \mathbf{U}_B \boldsymbol{\Sigma}_B \mathbf{V}_B^\top) \\
= &- \eta \left( \mathbf{V}_B \boldsymbol{\Sigma}_B \mathbf{U}_B^\top \mathbf{e} \mathbf{x}^\top \mathbf{V}_A \boldsymbol{\Sigma}_A^{-1} \mathbf{U}_A^\top + \mathbf{U}_A \boldsymbol{\Sigma}_A^{-1} \mathbf{V}_A^\top \mathbf{x} \mathbf{e}^\top \mathbf{U}_B \boldsymbol{\Sigma}_B \mathbf{V}_B^\top \right)
\end{aligned}$$

Combining the previous results:

$$\mathbf{U}_A = \mathbf{V}_B,$$
$$\boldsymbol{\Sigma}_A = \sqrt{\mu_1}\,\boldsymbol{\Sigma}_B,$$

and substituting into $\mathrm{d}(\mathbf{A}\mathbf{A}^\top)$, we obtain

$$\mathrm{d}(\mathbf{A}\mathbf{A}^\top) \approx \mu_1 \, \mathrm{d}(\mathbf{B}^\top\mathbf{B}).$$

Thus, we have established that the proportionality constants satisfy:

$$\mu_1 \approx \mu_2.$$

This result ensures that the scaled gradient descent method maintains balanced updates between $\mathbf{A}$ and $\mathbf{B}$, promoting stable training dynamics. $\qquad\square$

## C  EXPERIMENTAL DETAILS AND ADDITIONAL EXPERIMENTS

### C.1  EXPERIMENTAL DETAILS AND ADDITIONAL EXPERIMENTS OF GPT2 FINE-TUNING

In this section, we provide a detailed description of the experimental settings and additional experiments conducted for the fine-tuning of the GPT-2 model. First, in Appendix C.1.1, we present the experimental details of GPT2 fine-tuning, outlining the methodologies, datasets, and hyperparameters used to fine-tune GPT-2. Then, in Appendix C.1.2, we compare the performance of our proposed method $\pi$LoRA with LoRA+. The results demonstrate that $\pi$LoRA consistently outperforms LoRA+ across various evaluation metrics, highlighting the superior effectiveness of our approach.

### C.1.1 Experimental Details of GPT2 Fine-Tuning

In this section, we introduce the experimental settings for GPT-2. We strictly follow the same settings from the original LoRA (Hu et al., 2022) and Preconditioned LoRA (Zhang and Pilanci, 2024). We use the medium-size GPT-2 model (Radford et al., 2019), with hyperparameters listed in Table 3. The learning rates for iLoRA and $\pi$LoRA are individually tuned via grid search over the range $1 \times 10^{-4}$, $2 \times 10^{-4}$, ..., $9 \times 10^{-4}$, $1 \times 10^{-3}$, while the settings for LoRA and Preconditioned LoRA follow the default values from Zhang and Pilanci (2024). We train for 5 epochs using a linear learning rate schedule. It is worth noting that the AdamW hyperparameters $\beta_1$ and $\beta_2$ also follow the default values from Zhang and Pilanci (2024).

Table 3: Hyperparameters for GPT-2 fine-tuning on E2E

| Method | iLoRA | $\pi$LoRA |
|---|---|---|
| **Training** | | |
| Weight Decay | 0.01 | 0.01 |
| Dropout Probability | 0.1 | 0.1 |
| Batch Size | 8 | 8 |
| # Epochs | 5 | 5 |
| Warmup Steps | 500 | 500 |
| LR Scheduler | Linear | Linear |
| Label Smoothing | 0.1 | 0.1 |
| Learning Rate ($\times 10^{-4}$) | 6 | 7 |
| $\lambda$ | 10 | 1 |
| AdamW $\beta_1$ | 0.9 | 0.7 |
| AdamW $\beta_2$ | 0.999 | 0.8 |
| LoRA $\alpha$ | 32 | 32 |
| **Inference** | | |
| Beam Size | 10 | 10 |
| Length Penalty | 0.8 | 0.8 |
| No Repeat N-gram Size | 4 | 4 |

### C.1.2 Additional Experiments of GPT2 Fine-Tuning for LoRA+

In this section, we compare the performance of LoRA, LoRA+, and $\pi$LoRA on the E2E task using the GPT-2 model. Table 4 presents the experimental results across five evaluation metrics. We observed that $\pi$LoRA consistently outperforms both LoRA and LoRA+ across all metrics. While LoRA+ shows slight improvements over LoRA, $\pi$LoRA demonstrates the most significant gains, particularly in BLEU and NIST, solidifying its effectiveness in fine-tuning GPT-2 for the E2E task.

Table 4: Performance comparison of GPT-2 fine-tuning on E2E task: LoRA, LoRA+, and $\pi$LoRA.

| Method | Rank | E2E | | | | |
|---|---|---|---|---|---|---|
| | | BLEU | NIST | MET | ROUGE-L | CIDEr |
| LoRA | 4 | 68.9 | 8.69 | 46.5 | 71.4 | 2.51 |
| Lora + | 4 | 70.3 | 8.84 | 46.7 | 71.9 | **2.54** |
| $\pi$LoRA | 4 | **70.8** | **8.89** | **46.8** | **72.1** | **2.54** |

### C.2 Experimental Details and Additional Experiments of Mistral 7B Fine-Tuning

In this section, we provide a comprehensive overview of the experimental settings and additional experiments conducted for the fine-tuning of the Mistral 7B model. First, in Appendix C.2.1, we describe the experimental details of Mistral 7B fine-tuning, outlining the methodologies, datasets, and hyperparameters used throughout the experiments. Next, in Appendix C.2.2, we compare the

performance of our method, $\pi$LoRA, against LoRA+, demonstrating that $\pi$LoRA outperforms LoRA+ across various tasks. We also include additional experiments of Mistral 7B fine-tuning for 4-bit quantization in Appendix C.2.3, where we assess the effects of quantizing the model on performance and efficiency. Finally, in Appendix C.2.4, we compare the outcomes of experiments where the learning rate is not tuned (fixed learning rate) to those where the learning rate is tuned, demonstrating the robustness of our method to the learning rate.

### C.2.1 EXPERIMENTAL DETAILS OF MISTRAL 7B FINE-TUNING

In this section, we introduce the experimental settings for Mistral 7B. We follow the setting as in Zhang and Pilanci (2024), where LoRA factors are injected into each linear layer with a rank of $r = 16$. We trained for a total of 5 epochs with a batch size of 8. Apart from batch size, training epochs, and optimizer-related settings, the learning rate scheduler, warmup steps, warmup ratios, and maximum gradient norm remained at their default settings in the HuggingFace trainer class. The weight decay value was set to 0.01. For the five smaller tasks, MRPC, CoLA, RTE, STS-B, and WNLI, we used 1 * NVIDIA A100 GPU for training. For the other four larger tasks, we used 4 * NVIDIA A100 GPUs for training. For all tasks, we tuned the learning rate through grid search, specifically, for the six tasks (SST-2, MRPC, CoLA, RTE, STS-B, and WNLI), the range was $1 \times 10^{-5}$, $2 \times 10^{-5}$, ..., $9 \times 10^{-5}$, $1 \times 10^{-4}$, and for the other three tasks, the range was $1 \times 10^{-6}$, $2 \times 10^{-6}$, ..., $9 \times 10^{-6}$, $1 \times 10^{-5}$. We also performed grid search tuning for the regularization hyperparameter $\lambda$ over the range $1 \times 10^{-3}$, $1 \times 10^{-2}$, ..., $1 \times 10^{1}$, $1 \times 10^{2}$ and scaled regularization hyperparameter $\lambda$ over the range $1 \times 10^{-4}$, $1 \times 10^{-3}$, ..., $1 \times 10^{2}$, $1 \times 10^{3}$. In the experiments detailed in Appendix C.2.4, we verified that not tuning the learning rate or regularization hyperparameters resulted in only a minor performance drop in our method, which does not fundamentally affect the conclusions. The learning rate and regularization hyperparameters for each task are shown in Table 5, and other hyperparameters are listed in Table 6.

Table 5: Learning rate and regularization hyperparameter for Mistral 7B fine-tuning on GLUE. Scaled Reg is a hyperparameter introduced by Zhang and Pilanci (2024).

|  | MNLI | SST-2 | MRPC | CoLA | QNLI | QQP | RTE | STS-B | WNLI |
|---|---|---|---|---|---|---|---|---|---|
| iLoRA LR | 4.00E-06 | 1.00E-04 | 5.00E-05 | 5.00E-05 | 5.00E-06 | 8.00E-06 | 7.00E-05 | 1.00E-04 | 5.00E-05 |
| iLoRA $\lambda$ | 1 | 1 | 10 | 0.1 | 100 | 0.01 | 0.01 | 0.01 | 100 |
| $\pi$LoRA LR | 4.00E-06 | 6.00E-05 | 7.00E-05 | 8.00E-05 | 4.00E-06 | 8.00E-06 | 7.00E-05 | 8.00E-05 | 3.00E-05 |
| $\pi$LoRA $\lambda$ | 0.01 | 0.001 | 0.01 | 10 | 100 | 10 | 10 | 1 | 100 |
| $\pi$LoRA Scaled Reg | 0.001 | 0.01 | 1000 | 0.01 | 0.0001 | 0.001 | 10 | 0.1 | 1 |

Table 6: Other Hyperparameters for Mistral 7B Fine-Tuning on GLUE.

| Method | iLoRA& $\pi$LoRA |
|---|---|
| Train batch size | 8 |
| Seed (default) | 42 |
| AdamW $(\beta_1, \beta_2)$ | (0.9, 0.999) |
| AdamW $\epsilon$ | $1e^{-6}$ |
| LR Scheduler | linear |
| Num Epochs | 5 |
| Warmup steps & Warmup ratios | 0 |
| Weight decay | 0.01 |
| Max grad norm | 1 |
| LoRA rank | 16 |
| LoRA $\alpha$ | 16 |
| LoRA dropout | 0.05 |

### C.2.2 ADDITIONAL EXPERIMENTS OF MISTRAL 7B FINE-TUNING FOR LORA+

In this section, we compare the performance of $\pi$LoRA and LoRA+ on the GLUE benchmark tasks. Table 7 presents the experimental results. We find that $\pi$LoRA achieved the best overall

performance across all tasks, particularly excelling in MRPC and RTE with improvements of $1.96\%$ and $1.45\%$ respectively. While LoRA+ shows a slight advantage on the STS-B task, $\pi$LoRA demonstrates more consistent gains across different tasks. On average, $\pi$LoRA improves performance by $1.90\%$ compared to LoRA+, confirming its effectiveness in a variety of scenarios.

Table 7: Performance Comparison between $\pi$LoRA and LoRA+ on GLUE Benchmark.

| Method | Rank | GLUE | | | | | | | Avg. |
|---|---|---|---|---|---|---|---|---|---|
| | | SST-2 | MRPC | CoLA | QNLI | RTE | STS-B | WNLI | |
| LoRA | 16 | 96.79 | 88.48 | 71.05 | 94.42 | 90.61 | 90.42 | 81.69 | 87.64 |
| LoRA+ | 16 | 96.90 | 88.48 | 70.90 | 95.22 | 90.25 | **92.50** | 80.28 | 87.79 |
| $\pi$LoRA | 16 | **97.25** | **90.44** | **71.97** | **95.37** | **91.70** | 92.35 | **88.73** | **89.69** |

### C.2.3 ADDITIONAL EXPERIMENTS OF MISTRAL 7B FINE-TUNING FOR 4BIT QUANTIZATION

In the main body of the paper, we present results without applying 4-bit quantization to Mistral 7B while in Zhang and Pilanci (2024) quantization is applied. Here, we experimentally verified that 4-bit quantization had little effect on the experimental results and the proposed methods can still outperform the baselines. In this section, we compare the impact of using 4-bit quantization versus not using it on iLoRA in the GLUE benchmark tasks. Table 8 presents the experimental results. We found that 4-bit quantization has minimal impact on model performance. For iLoRA, the 4-bit version slightly outperforms the non-quantized version in most tasks, but the differences are marginal. This indicates that 4-bit quantization can improve memory and computational efficiency while maintaining comparable model performance.

Table 8: Comparison of 4-bit Quantization Impact on iLoRA on GLUE Benchmark.

| Method | 4bit | Rank | GLUE | | | | | Avg. |
|---|---|---|---|---|---|---|---|---|
| | | | MRPC | CoLA | RTE | STS-B | WNLI | |
| LoRA | Y | 16 | 88.48 | 71.05 | 90.61 | 90.42 | 81.69 | 84.45 |
| Preconditioned LoRA | Y | 16 | 89.46 | 71.30 | 91.34 | 91.10 | 83.10 | 85.26 |
| iLoRA | N | 16 | 89.71 | 71.90 | 90.98 | **92.25** | **87.32** | 86.43 |
| iLoRA | Y | 16 | **90.93** | **72.51** | **92.06** | 92.24 | 85.92 | **86.73** |

### C.2.4 ADDITIONAL EXPERIMENTS OF MISTRAL 7B FINE-TUNING FOR FIXED LEARNING RATE

In this section, we provide a comprehensive comparison of LoRA, iLoRA, and variations of iLoRA with fixed learning rate and fixed regularization hyperparameter on the GLUE benchmark tasks using the Mistral 7B model. Table 9 presents the results of experiments comparing LoRA, iLoRA, and iLoRA with a fixed learning rate. For the five smaller tasks (WNLI, STS-B, RTE, MRPC, and CoLA), the learning rate was fixed at $5e^{-5}$, while for the other four tasks, it was fixed at $1e^{-5}$. We found that the performance loss of the fixed learning rate version of iLoRA is minimal and remains highly competitive. Additionally, Table 10 highlights the performance comparison between LoRA, iLoRA, and iLoRA with a fixed regularization hyperparameter ($\lambda = 0.1$) across five smaller tasks (WNLI, STS-B, RTE, MRPC, and CoLA). Here, iLoRA consistently achieves the highest scores, while the fixed regularization version also performs strongly. These results emphasize the flexibility and effectiveness of iLoRA in various experimental settings, confirming its robustness in both learning rate and regularization parameter configurations.

Table 9: Comparison of LoRA, iLoRA, and iLoRA with fixed learning rate on GLUE benchmark for Mistral 7B.

| Method | Rank | GLUE | | | | | | | | | Avg. |
|---|---|---|---|---|---|---|---|---|---|---|---|
| | | MNLI | SST-2 | MRPC | CoLA | QNLI | QQP | RTE | STS-B | WNLI | |
| LoRA | 16 | 89.86 | 96.79 | 88.48 | 71.05 | 94.42 | 91.24 | 90.61 | 90.42 | 81.69 | 88.28 |
| iLoRA | 16 | **91.59** | **97.13** | **89.71** | **71.90** | **95.20** | **91.43** | **90.98** | **92.25** | **87.32** | **89.72** |
| iLoRA(Fixed LR) | 16 | 91.17 | 97.02 | **89.71** | **71.90** | 94.86 | 91.37 | 89.89 | 91.84 | **87.32** | 89.45 |

Table 10: Comparison of LoRA, iLoRA, and iLoRA with fixed regularization Hyperparameter on GLUE benchmark (five small tasks).

| Method | Rank | GLUE | | | | | Avg. |
|---|---|---|---|---|---|---|---|
| | | MRPC | CoLA | RTE | STS-B | WNLI | |
| LoRA | 16 | 88.48 | 71.05 | 90.61 | 90.42 | 81.69 | 84.45 |
| iLoRA | 16 | **89.71** | **71.90** | **90.98** | **92.25** | **87.32** | **86.43** |
| iLoRA(Fixed $\lambda$) | 16 | 88.97 | **71.90** | 90.61 | 92.04 | 85.92 | 85.89 |

## C.3 EXPERIMENTAL DETAILS AND ADDITIONAL EXPERIMENTS OF DIFFUSION MODEL FINE-TUNING

### C.3.1 EXPERIMENTAL DETAILS OF DIFFUSION MODEL FINE-TUNING

In diffusion model experiments, we based our work on the Mix-of-Show model (Gu et al., 2023) repository. We followed the default settings of (Gu et al., 2023) but made modifications according to those in Zhang and Pilanci (2024). We used Chilloutmix[2] as the pre-trained model, and the rank of LoRA was set to 4. For sampling, we chose DMP-Solver (Lu et al., 2022). For more details on the experimental setup, please refer to (Gu et al., 2023). In the experiment described in Section 6.2, we fixed the learning rate of the text embedding to $1 \times 10^{-3}$ and used different learning rates for the text encoder and UNet. For experimental results with different LoRA parameter fusion coefficients and various prompts, please refer to Appendix C.3.2.

### C.3.2 ADDITIONAL EXPERIMENTS DIFFUSION MODEL FINE-TUNING

First, we conduct experiments to test the LoRA and iLoRA under different fusion coefficients. The experimental setup is the same as Fig. 1, with learning rates chosen as $(5e - 4, 5e - 4)$. Fig. 4 shows the experimental results. The first row has a LoRA parameter fusion coefficient of $0.7$, and the second row is $1$. The first two columns are results generated by LoRA, and the last two columns are results generated by iLoRA. It can be seen that iLoRA produces higher quality images, and in some images, LoRA ignores the keyword "eiffel tower".

In the second experiment, we tested the results of LoRA, iLoRA, and $\pi$LoRA on a new prompt: "a pencil sketch of $\langle V_{\text{potter}} \rangle$". We used the same experimental settings as in Fig. 1, only changing the prompt. The results are shown in Fig. 5. It can be seen that iLoRA and $\pi$LoRA generate images that are significantly better than those generated by LoRA.

---

[2]https://civitai.com/models/6424/chilloutmix

| Fusion Coefficient | LoRA | | iLoRA | |
|---|---|---|---|---|
| | Image 1 | Image 2 | Image 1 | Image 2 |

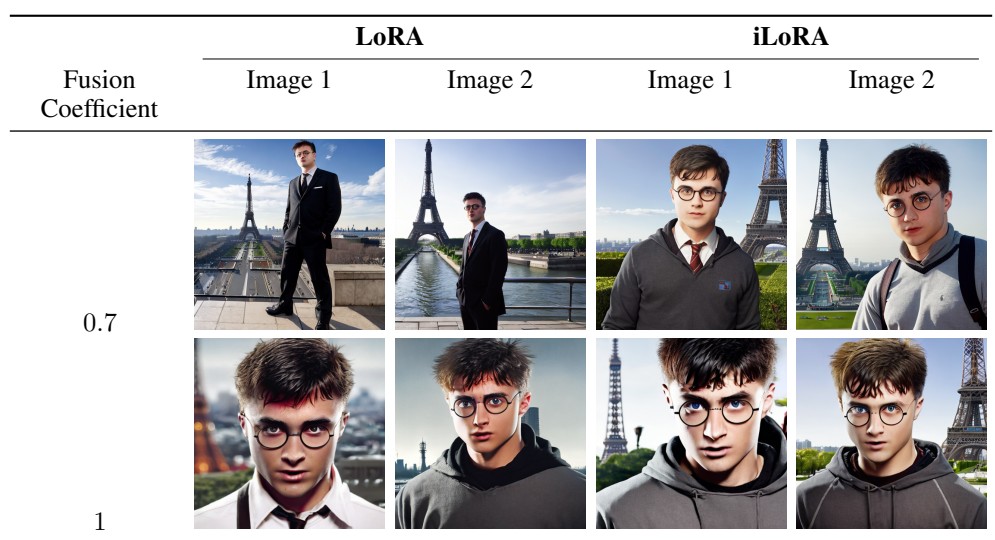

| 0.7 | | | | |
| 1 | | | | |

Figure 4: Comparison of images generated with LoRA and iLoRA across different fusion coefficient. The two rows correspond to different fusion coefficients: 0.7 in the first row and 1 in the second row. The first two columns show results from LoRA, and the last two columns show results from iLoRA.

| Learning Rates | LoRA | | iLoRA | | $\pi$LoRA | |
|---|---|---|---|---|---|---|
| | Image 1 | Image 2 | Image 1 | Image 2 | Image 1 | Image 2 |

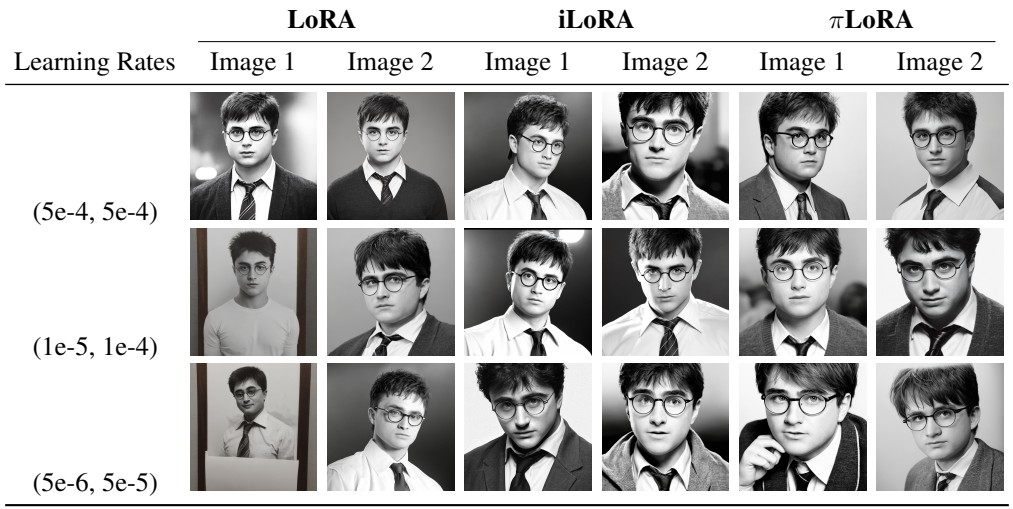

| (5e-4, 5e-4) | | | | | | |
| (1e-5, 1e-4) | | | | | | |
| (5e-6, 5e-5) | | | | | | |

Figure 5: Comparison of images generated with LoRA, iLoRA, and $\pi$LoRA across different learning rates for the Mix-of-Show model. The three rows correspond to three different sets of learning rates for (text encoders, U-Nets): (5e-4, 5e-4), (1e-5, 1e-4), and (5e-6, 5e-5). The first and second columns show results from LoRA, the third and fourth columns show results from iLoRA, and the fifth and sixth columns show results from $\pi$LoRA.

