# OpenReview forum: "Imbalance-Regularized LoRA: A Plug-and-Play Method for Improving Fine-Tuning of Foundation Models"
_ICLR.cc/2025/Conference — Submitted to ICLR 2025_

### Official Review · Reviewer_vaV8 · 2024-10-29

**Soundness:** 1
**Presentation:** 2
**Contribution:** 1
**Rating:** 3
**Confidence:** 4

**Summary:**

This paper proposes a novel variant of LoRA, termed *imbalance-regularized LoRA*. Based on the observation that matrices $A$ and $B$ are inherently asymmetrical, it introduces a regularization term with a constant scaling factor solely determined by the dimensions of the original weight matrix and LoRA rank. This scaling factor is neither trainable nor tunable, adding no additional complexity for the user. Furthermore, to maintain imbalance during backpropagation, the paper also introduces a constant scaling factor to adjust the learning rate. Empirical results indicate that the proposed method outperforms LoRA in fine-tuning both large language models (LLMs) and diffusion models.

**Strengths:**

This paper proposes a novel variant of LoRA, termed “imbalance-regularized LoRA”. Based on the observation that matrices $A$ and $B$ are inherently asymmetrical, it introduces a regularization term with a constant scaling factor solely determined by the dimensions of the original weight matrix and LoRA rank. This scaling factor is neither trainable nor tunable, adding no additional complexity for the user. Empirical results indicate that the proposed method outperforms LoRA in fine-tuning both large language models (LLMs) and diffusion models.

**Weaknesses:**

This paper should be rejected due to several fundamental flaws:

1. The theoretical analyses rely on unrealistic assumptions, casting doubt on whether they are applicable to generic cases. In Section 4.1, the paper assumes that the variance of $A x$ and $B A x$ remains constant. It further assumes that $A A^\top = \mu_1 B^\top B$, which is improbable in practical scenarios. The same assumption applies in Section 4.2.

2. The analysis in Section 4.2 addresses only a single layer, which is not representative of modern LLMs or diffusion models with multi-layer architectures. Points 1 and 2 may therefore render the paper’s theoretical contributions negligible.

3. The empirical results are only compared to vanilla LoRA and do not include other LoRA variants. A comparison with LoRA+, which also addresses asymmetry between $A$ and $B$, would be useful, especially as LoRA+ is mentioned in the related works section.

4. Figure 3 appears to have been cut off prematurely at $r / m = 2$. In both charts, $r / m = 2$ yields significantly better results than $r / m = 1.5$. If this trend continues, larger $r / m$ ratios could demonstrate even better results than those achieved at $r / m = 1$, suggesting a possible preference for larger $r / m$ in networks with multiple layers. This limitation echoes the unrealistic assumptions noted in points 1 and 2 above.

5. The experiments did not examine key hyperparameters that could impact results substantially. At a minimum, the learning rate and dropout rate should be varied to show stability across different settings.

6. Due to the omission noted in point 5, it is unclear whether the effect of this new regularization term could be partially reproduced by other regularizations, such as dropout, which are simpler and computationally cheaper. Points 3 through 6 may ultimately undermine the empirical contributions of this paper.

**Questions:**

1. Does the theoretical analysis hold without assuming that the variance of $A x$ and $B A x$ remain constant?
2. Does it hold without assuming $A A^\top = \mu_1 B^\top B$?
3. Does the analysis apply to models with multiple layers?
4. How does this method compare to LoRA+?
5. How does it perform when $r/m > 2$?
6. How does it perform under different learning and dropout rates?
7. How does it compare to dropout, and does the gain justify the additional computation cost?

---

### Official Review · Reviewer_8NxG · 2024-11-04

**Soundness:** 2
**Presentation:** 3
**Contribution:** 2
**Rating:** 5
**Confidence:** 4

**Summary:**

In this paper, the authors aim to address the problem brought by the asymmetry between matrices A and B in LoRA. To achieve this goal, the authors proposed iLoRA to incorporate a regularization term into LoRA. The proposed iLoRA is plug-and-play. Furthermore, the authors combine iLoRA with Riemannian Preconditioned LoRA to proposed \piLoRA to deliver significant performance gains to iLoRA across diverse tasks.

**Strengths:**

(1)The challenge of inherent asymmetry between A and B in LoRA is interesting.

(2)The paper is well-written and easy to follow.

**Weaknesses:**

(1) The author claims that they try to address and challenge brought by the asymmetry between A and B in LoRA. However, the authors did not provide any experimental analysis to showcase how the asymmetry will affect the training of LoRA.

(2) The experiments are not comprehensive, the authors did not include mainstream LLMs, like Llama-7B and Llama-13B.

(3) The experiments on NLG tasks should be provided.

(4) The authors claimed that iLoRA can enhance the stability of LoRA. However, no analysis on training stability is provided to support this claim.

(5) The contribution of \piLoRA is incremental.

**Questions:**

Please see the weaknesses.

---

### Official Review · Reviewer_a7jc · 2024-11-04

**Soundness:** 4
**Presentation:** 4
**Contribution:** 3
**Rating:** 6
**Confidence:** 2

**Summary:**

This paper introduces Imbalance-Regularized LoRA (iLoRA), a novel method aimed at addressing the inherent asymmetry between matrices A and B in LoRA-based understanding, ultimately enhancing training stability and performance.

**Strengths:**

- The paper is logically structured and clearly articulated, with a well-organized mathematical exposition. The resulting design aligns well with the stated objectives.
- The experiments are conducted thoroughly, showing significant performance improvements in fine-tuning experiments on both GPT-2 and MISTRAL 7B.
- The proposed method is easy to apply, requiring only minimal modifications to existing LoRA code. It is an elegant, clean, and interpretable optimization with minimal additional overhead.

**Weaknesses:**

1. It’s unclear how significant the performance improvement is in the diffusion section, and why quantitative metrics like FID and CLIP score were not included.
2. A key advantage of LoRA is its resistance to overfitting, which offers advantages in certain out-of-domain transfer learning tasks, such as Alpaca and Instruct-Eva. Demonstrating improvements in these tasks could greatly enhance the impact of this paper.
3.It be possible to expand on the reasoning in the introduction regarding how addressing the inherent asymmetry between matrices A and B improves training stability and performance? This would make the paper more accessible to readers and clarify its practical significance.

**Questions:**

NA

---

> ### Comment · Reviewer_a7jc · 2024-11-30
>
> I still cannot fully buy in the theoretical contributions of this paper, particularly as I don't believe it fundamentally advances the development of PEFT or LoRA. In terms of performance, I find the experimental results in this paper to be acceptable. I would rate this work around a 5-6 at ICLR, marginally above the acceptance threshold.

---

### Comment · Area_Chair_fyhH · 2024-11-24
**Reminder - Public Discussion Phase Ending Soon**

Dear PC memebers,

Thank you for your valuable comments during the review period, which raised many interesting and insightful questions. Now the discussion period is coming to a close, please take a moment to review the authors’ responses if you haven’t done so already. Even if you decide not to update your evaluation, kindly confirm that you have reviewed the responses and that they do not change your assessment.

Timeline: As a reminder, the review timeline is as follows:

November 26: Last day for reviewers to ask questions to authors.
November 27: Last day for authors to respond to reviewers.
November 28 - December 10: Reviewer and area chair discussion phase.
December 20: Meta reviews and initial decisions are due.


Thank you for your time and effort!

Best regards,
AC

---

### Meta-Review · Area_Chair_fyhH · 2024-12-18

**Metareview:**

This paper proposes a novel regularization approach to address the asymmetry between A and B in LoRA. The idea is innovative and highlights a potentially important direction for improving LoRA. However, the current manuscript, along with the subsequent discussions, fails to sufficiently convince the reviewers or myself of the validity and effectiveness of this approach. Therefore, I believe this paper should be rejected at this stage to ensure theoretical rigor. I encourage the authors to continue their work in this promising direction and hope they can eventually contribute a fundamental improvement to the field.

**Additional Comments On Reviewer Discussion:**

The topic on the asymetry between A and B in LoRA is interesting and hence has active discussion. These insightful exchanges have been helpful for both the authors and myself in evaluating the paper. Notably, Reviewer vaV8 (score = 3) raised several thoughtful points during the discussion with the authors. Even the sole positive reviewer (a7jc, score = 6) expressed reservations about fully endorsing the theoretical contributions, suggesting that the actual value lies closer to 5-6. Overall, the concerns regarding the theoretical aspects of the paper are consistent across the reviews.

---

### Decision · Program_Chairs · 2025-01-22

Reject